# Chromatin heterogeneity modulates nuclear condensate dynamics and phase behavior

Jing Xia[1], Jessica Z. Zhao [1], Amy R. Strom [1] & Clifford P. Brangwynne [1,2,3,4] ✉

The cell nucleus is a soft composite material with a shell-like nuclear cortex enclosing chromatin, comprised of roughly 2 meters of DNA and associated proteins. Assembling on and around chromatin are droplet-like structures known as biomolecular condensates, which form via phase separation, and facilitate vital roles in gene expression. From studies in non-living materials, the driving forces for phase separation are expected to be sensitive to the local mechanical environment, which often exhibits significant spatial heterogeneity. However, the relationship between chromatin heterogeneity and the phase equilibrium and dynamics of nuclear condensates remains unclear. Here, we investigate the interplay between chromatin organization and the formation, dynamics, and size of engineered model condensates and endogenous nuclear bodies in living cells. We demonstrate that decreasing chromatin heterogeneity with epigenetic modifying drugs correlates with decreased mobility of both endogenous and engineered condensates, and is associated with impaired condensate growth and shifts in the binodal phase boundary of engineered condensates. These findings illustrate how the cell nucleus behaves as a heterogeneous composite material with mechanically permissive chromatin micro-environments.

The nucleus is the defining feature of eukaryotic cells, which evolved to compartmentalize the genetic material and regulate gene expression. The nucleus exhibits a complex composition and dynamic organization[1,2], which is linked to essential functions including DNA replication and repair, epigenetic regulation, as well as RNA transcription, splicing and modifications. Inside the nucleus, chromatin serves as a scaffold for these dynamic activities while various nuclear bodies or condensates within the nucleoplasm[3], such as nucleoli[4], Cajal bodies[5], and nuclear speckles[6], facilitate various steps of RNA processing and ribonucleoprotein assembly.

In addition to the rich biochemical activities, the nucleus shows complex material behavior. The mechanical properties of the nucleus are predominantly influenced by two major contributors: chromatin and lamin proteins, both of which exhibit spatial heterogeneity. Chromatin, consisting of DNA and associated proteins, may contribute to the overall stiffness and viscoelastic behavior of the nucleus[7,8]. Its

compaction levels vary across genomic regions, influencing essential processes on and around the genome[9,10]. Compacted chromatin regions marked by H3K9me[2,3] restrict access to DNA, resulting in gene silencing, while open-chromatin regions marked by H3K9ac promote gene expression by enhancing DNA accessibility[11]. Additionally, lamin proteins, which constitute the nuclear lamina, contribute to nuclear mechanics by providing overall structural support and stability[12,13]. While chromatin spans the majority of the nucleus, the nuclear lamina mostly resides on the inner surface of the nuclear envelope. The difference in spatial distribution likely influences their distinct roles in overall nuclear mechanics, with the lamina primarily affecting the cortical region while chromatin largely dictates the mechanical properties of the nuclear interior[14,15]. The role of lamin in nuclear mechanics has been extensively investigated owing to its relatively simpler structure, but the contribution of chromatin remains less clear. This is primarily due to the significant spatial heterogeneity of chromatin

[1]Department of Chemical and Biological Engineering, Princeton University, Princeton, NJ, USA. [2]Omenn-Darling Bioengineering Institute, Princeton University, Princeton, NJ, USA. [3]Princeton Materials Institute, Princeton University, Princeton, NJ, USA. [4]Howard Hughes Medical Institute, Princeton University, Princeton, NJ, USA. ✉e-mail: cbrangwy@princeton.edu

throughout the nucleus, coupled with its complex hierarchical organization spanning length scales from nanometers to micrometers.

This spatial complexity of the chromatin material is intimately coupled to gene expression programs, with undifferentiated stem cells exhibiting more uniform chromatin, and heterogeneity increasing as cells become progressively more committed to particular cell fates[16,17]. Moreover, a number of studies have directly tied mechanical heterogeneity of the nucleus and local mechanical cues to the execution of specific genetic programs[18–20]. There is also growing evidence suggesting that alterations in the heterogeneity of nuclear structure are linked to a variety of disease states[21–27]. For example, dysregulated nuclear structure and disruptions in chromatin distribution are frequently observed in cancer cells and with neurodegenerative diseases such as Alzheimer's disease, Parkinson's disease, and Huntington's disease[24–26], as well as in laminopathies such as the premature aging disease progeria (HGPS)[27].

The dozens of different biomolecular condensates within the nucleus are closely linked to the genetic material, as they form directly on and around chromatin[6,28–30]. This interplay with the heterogeneous chromatin is particularly interesting, given that condensates are known to form through phase separation and related phase transitions[31–33]. For example, specific genomic loci may act as nucleation sites that promote localized phase separation of liquid transcriptional condensates[34–38], while other condensates tend to grow in regions with low chromatin density[39].

A number of recent studies using nonliving systems suggest that the mechanical properties of the condensate environment could influence the phase equilibrium of condensates[40–48]. In particular, mechanical heterogeneity of the substrate has been shown to affect phase separation, both in vitro[40,41,49] and theoretically[48]. Consistent with these studies, condensates in living cells are increasingly recognized as interacting mechanically with the intracellular environment[39,46,50–52], suggesting that condensate phase equilibria could potentially be mechanically modulated in living cells. For example, in the cell nucleus, the mechanics of the chromatin network has been suggested to influence the dynamics of condensates formed within it[50,53]. However, it remains unclear whether mechanically-driven phase shifts occur in cells, and particularly how the high degree of spatial heterogeneity of the nuclear material would impact local condensate dynamics and phase behavior.

In this study, we show that altering chromatin spatial heterogeneity can significantly impact condensate phase behavior. Decompaction of the chromatin network results in reduced nuclear heterogeneity, which strongly correlates with impaired growth rates, reduced sizes, and decreased mobility of embedded condensates following phase separation. These changes appear to result from homogenization of the chromatin network directly inhibiting phase separation, as measured by shifts in the binodal phase boundary toward a higher concentration. By contrast, relaxing constraints of the chromatin network at the nuclear boundary by knocking down Lamin proteins does not significantly impact overall phase equilibrium, suggesting the nuclear cortex does not impact local condensate equilibria. These findings underscore the major effects of manipulating heterogeneity within the chromatin network on condensate formation and dynamics within the nucleus, shedding light on the intricate interplay between chromatin organization and the emergent properties of nuclear condensates.

## Results

### Chromatin decompaction reduces nuclear heterogeneity

To first examine the chromatin distribution in living cells, we use lentiviral transduction to induce miRFP670-tagged (miRFP-) H2B construct into U2OS cells to label the chromatin network, followed by 3D confocal fluorescence imaging. We verified that the incorporation of miRFP670-H2B only caused a minimal increase (~5%) of the total amount of H2B present in the U2OS cells, as shown in the western blot quantification results (Supplementary Fig. 1a, b, c). We also verified that the incorporation of miRFP670-H2B does not change cell viability (Supplementary Fig. 2a, b), proliferation (Supplementary Fig. 3), or the cell cycle (Supplementary Fig. 4a, b). We confirmed that more than 97% of the expressed miRFP670-H2B is incorporated into the chromatin, as demonstrated by fluorescence recovery after photobleaching (FRAP) (Supplementary Fig. 5a, b). Therefore, the fluorescence signal from miRFP670-H2B accurately represents the chromatin distribution at the endogenous level.

This variation in the spatial density of the chromatin network, which we refer to as simply chromatin heterogeneity, can be measured and quantified using a confocal microscope. To modulate the global spatial heterogeneity of the chromatin network, we change the compaction state of the condensed chromatin network by utilizing Trichostatin A (TSA), a small molecule drug that inhibits histone deacetylases (HDACi) and therefore increases histone acetylation[54]. Increased acetylation of lysine residues on histones reduces their overall positive charges and therefore weakens interactions between histones and negatively charged DNA, resulting in decompaction of chromatin[55,56] (Fig. 1a). It has been widely documented that TSA leads to acetylation of normally hypoacetylated areas such as constitutive heterochromatin, which thus becomes decompacted upon TSA treatment[55,57–59]. Importantly, the increase in overall acetylation levels leads to an overall increase in the homogeneity of chromatin distribution across the nucleus[55,59–65]. As expected, chromatin decompaction entails chromatin fibers occupying a larger effective volume fraction within the nucleus, with an associated increase of the nuclear volume (Supplementary Fig. 6). Upon TSA treatment, chromatin clusters observed in wild-type (WT) cells mostly disappear, indicating that the chromatin network has become more homogeneous. Representative images comparing the same cell before and after overnight TSA treatment are depicted in Fig. 1b, where the fluorescence intensity of the miRFP670-H2B is color-coded.

The visually clear decrease in spatial heterogeneity upon TSA treatment can be quantified by calculating a coefficient of variation (COV), defined as the ratio of the standard deviation of fluorescence intensity to the average fluorescence intensity, with background intensity subtracted from miRFP670-H2B. There is a consistent decline in the coefficient of variation (COV) with increasing duration of TSA treatment, indicating a progressive decrease in chromatin heterogeneity over time (Fig. 1c). Furthermore, our 3D quantification of chromatin heterogeneity shows a consistent decrease in heterogeneity with longer TSA treatment durations (Supplementary Fig. 7). To confirm the generality of these findings, we also introduced a miRFP670-tagged (miRFP-) H2B construct into a different cell line, HeLa, and confirmed the nearly complete incorporation of miRFP670-H2B (Supplementary Fig. 5c, d), with minimal disturbance to total H2B expression levels (Supplementary Fig. 1d, e, f) and low toxicity to the cells following lentiviral transduction (Supplementary Fig. 2c, d). Consistently, we also observed a decrease in chromatin heterogeneity when HeLa cells were treated with TSA (Supplementary Fig. 8a, b). This finding suggests that TSA reduces the heterogeneity of the chromatin network independently of the cell line, consistent with other studies[55,59–65].

To gain insight into the impact of these changes on mesoscale chromatin heterogeneity on the local microenvironment, we introduce genetically encoded multimeric (GEM) nanoparticles GEM40-NLS-mGFP into the cell, co-expressed with miRFP670-H2B to visualize chromatin, and conduct imaging-based particle tracking. The GEM40 nanoparticles have an average diameter of 40 nm, which is on the same order of magnitude as the typical mesh size of the chromatin network[66–69]. The dynamics of GEM40 nanoparticles provide a qualitative assessment of the local chromatin mesh size relative to their own size, reflecting mesoscale structural changes and chromatin mechanics. In low-density chromatin regions, GEM40 nanoparticles move

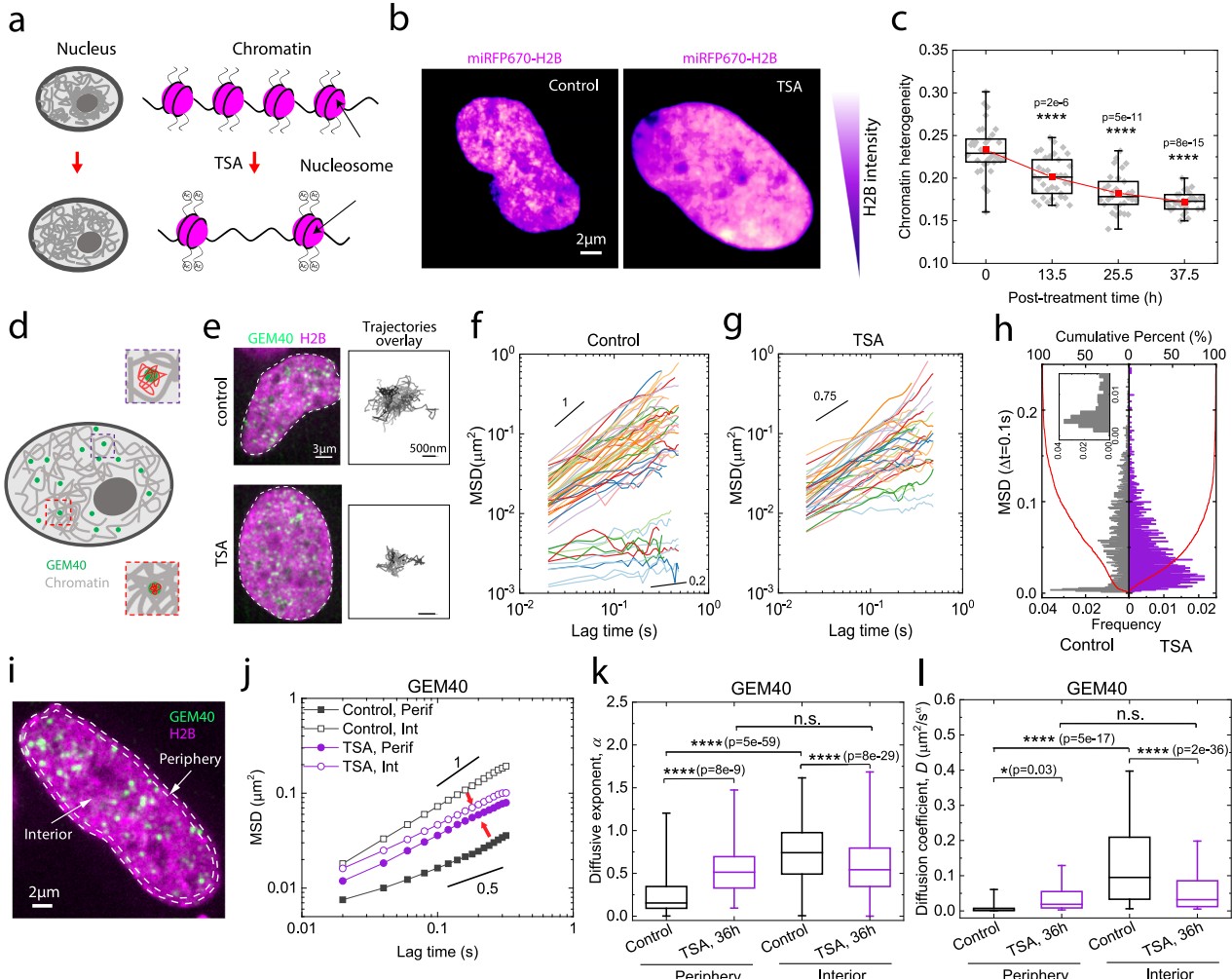

**Fig. 1 | Tuning chromatin heterogeneity of U2OS cells by decompacting the chromatin network with TSA. a** Mechanism of chromatin decompaction by Trichostatin A (TSA). **b** Heatmap of fluorescence intensity of miRFP670-H2B tagged nuclei before and after TSA treatment. **c** Chromatin heterogeneity of cells under TSA treatment. n = 34, 37, 31, 27 nuclei for each group, respectively. **d** Schematic of diffusing GEM40 within chromatin network in the nucleus. Dense and sparse chromatin regions are shown as zoom-in figures. GEM40 trajectories are marked in red. **e** Representative fluorescence image of GEM40 in miRFP670-H2B-tagged cells before and after TSA treatment. GEM trajectory overlays are shown to the right. Representative MSD of GEM40 in the cell before (**f**) and after (**g**) TSA treatment. Black solid lines are visual guides. **h** Histogram of MSD amplitude of GEM40 at lag time of 0.1 s. Red curves represent the cumulative distribution function. **i** Representative image of a nucleus with dashed contours delineating the interior and peripheral regions, overlaid with fluorescence images of the nucleus. **j** MSD of GEM40 in the nuclear interior and periphery of cells before and after TSA treatment. Black solid lines are visual guides. **k, l** Diffusive exponent and diffusion coefficient of GEM40 in the interior and periphery of cell nuclei before and after TSA treatment. For (**h, j, k, l**), n = 38 cells for control, and n = 35 cells for TSA. In the box plots, 25%, 50%, and 75% of the data are indicated using hinges and center lines. For (**c**) and (**k**), the whiskers extend from the hinges to the highest and lowest values. For (**l**), the whiskers extend from the hinges to the 10% and 90% of the data. In (**c, k, l**), asterisks indicate two-sided p values based on student's t-tests. p > 0.05 is considered non-significant. * for p < 0.05, ** for p < 0.01, *** for p < 0.001, and **** for p < 0.0001. For (**e, i**), the experiment was repeated three times independently with similar results. For (**c, h, j, k, l**), three biological replicates are used for statistical analysis.

freely and explore large areas, while in high-density regions, they become entrapped and explore less area (Fig. 1d). In TSA-treated cells, we observe a reduction in the explored area of GEM40 trajectories compared to untreated cells (Fig. 1e). We find that the mean square displacement (MSD) of GEM40 nanoparticles in untreated cells distinctly diverges into two populations: slower ones with diffusive exponent close to 0.5 and faster ones with diffusive exponent close to 1 (Fig. 1f). This behavior is also reflected in the bimodal distribution of MSD at lag time of 0.1 s (Fig. 1h). In contrast, the MSD of GEM40 nanoparticles in TSA-treated cells appears as a unified group with a diffusive exponent close to 0.75 (Fig. 1g), which can also be reflected by the unimodal distribution of MSD at a lag time of 0.1 s (Fig. 1h). These findings are consistent with TSA decreasing heterogeneity within the chromatin network, promoting a more uniform mesh size throughout

the whole nucleus, and thereby giving rise to a more unimodal distribution of GEM mobility.

We hypothesized that the bimodal distribution of GEM40 mobility arises from spatially varying distributions of heterochromatin and euchromatin. Heterochromatin, the more densely packed chromatin variant, mainly occupies the nuclear periphery and some regions around the edge of the nucleolus[70,71] To examine these spatial differences, we define two regions: nuclear periphery and nuclear interior, and measure the dynamics of GEM40 in those regions (Fig. 1i). Consistent with GEM40 mobility reflecting these density variations, we find that the mean square displacement (MSD) of GEM40 nanoparticles within the nuclear interior of untreated cells greatly exceeds that of GEM40 nanoparticles in the nuclear periphery. Moreover, after TSA treatment, there is a significant decrease in the MSD of GEM

nanoparticles within the nuclear interior, while the MSD of GEM40 nanoparticles within the nuclear periphery actually increases (Fig. 1j).

These changes in mobility of GEM40 nanoparticles before and after TSA treatment were quantified by analyzing their MSD using an anomalous diffusion model ($MSD = Dt^{\alpha}$), and used to compute the anomalous diffusion coefficient ($D$) and the anomalous diffusive exponent ($\alpha$). After the TSA treatment of 36 h, GEM40 at the nuclear periphery showed an increased diffusive exponent from 0.2 to 0.7, while those within the nuclear interior decreased from 0.7 to 0.6 (Fig. 1k). A similar trend is observed in the diffusion coefficient (Fig. 1l). Additionally, when considering all populations of GEM40 nanoparticles, the overall MSD exhibits an overall downward shift with prolonged TSA treatment (24 h and 36 h) (Supplementary Fig. 9a), as well as decreased diffusive exponent and coefficient (Supplementary Fig. 9b, c). These GEM40 results suggest that while chromatin-rich regions undergo decompaction, there is a concomitant densification of the normally chromatin-poor regions; homogenization appears to lead to an overall reduction in the mesh size of the chromatin network.

### Homogenized chromatin network gives rise to impaired condensate growth and sizes

To investigate the effect of chromatin homogenization on the formation of the condensates, we use the Corelet optogenetic system, an intracellular model reconstitution system which allows for mapping the binodal phase boundary of condensates in living cells[38,52,72]. The Corelet system consists of two modules: a "Core" made of 24 units of human ferritin heavy chain (FTH1) fused with an engineered protein iLID and a nuclear localization signal (NLS), and an intrinsically disordered region (IDR) unit fused to SspB. Upon blue light activation, iLID heterodimerizes with SspB, leading to condensate formation through IDR-driven phase separation (Fig. 2a). After introducing the Corelet system into U2OS cells, we image cells before and after blue light stimulation (Fig. 2b).

Upon exposure to blue light, condensates form within the nucleus, increasing in size over time (Fig. 2c, d). We find that condensates are clearly larger in untreated cells compared to TSA-treated cells (32 h), exhibiting a similarly shaped size distribution with shifted mean, regardless of 15 s activation (Fig. 2e, g) and 180 s activation (Fig. 2f, h). In addition, the condensate size consistently decreases in cells treated with TSA for longer durations (16 h, 24 h, and 32 h) (Supplementary Fig. 10a, b). Consistently, a decrease in condensate size is observed in HeLa cells treated with TSA, indicating that this phenomenon is not specific to a particular cell line (Supplementary Fig. 11a, b). To investigate whether the shift in the size of the condensates is related to the redistribution of the chromatin network density, we light-activate the condensates in cells transfected with miRFP670-H2B. We find that the condensates are generally excluded by the chromatin network (Supplementary Fig. 12a), consistent with previous findings[39]. We measure the condensate size after 3 min of activation and chromatin density before activation in the same regions, and calculate the Pearson correlation coefficient. We find condensate size and chromatin density are negatively correlated, which holds true for both untreated and TSA-treated cells (Supplementary Fig. 12b). This suggests that locally higher chromatin density universally hinders the growth of condensates, consistent with prior studies suggesting that the chromatin network imparts mechanical resistance[39,53].

To investigate the growth dynamics of condensates, we analyze the number and average size of condensates during the 3-min activation period. Condensate numbers initially increase within the first minute, reflecting nucleation and growth of condensates, followed by a plateau and slight decrease in the subsequent 2 min, reflecting the coarsening and coalescing of condensates[38,50]. Despite a continuous increase in condensate number and size during activation with and without TSA treatment, untreated cells have more and larger sizes of condensates (Fig. 2i, j). Scaling analysis of size growth during the

coarsening stage reveals a coarsening exponent of ~0.16 for untreated cells, with a smaller exponent of ~0.11 for TSA-treated cells. The exponent for untreated cells is approximating one-third of its diffusive exponent (0.55, Fig. 3c), consistent with expectations for Brownian-motion-driven coalescence[53], as frequently observed in Supplementary Movie 1. Interestingly, however, the coarsening exponent for TSA-treated cells was lower than expected, approximating one-fifth of its diffusive exponent of (0.5, Fig. 3c), as shown in Fig. 2k. A similarly decreased growth rate caused by TSA has recently been reported in a chemically induced phase separation system, aligning with our observations[73]. To investigate whether the size inhibition by chromatin homogenization can be recapitulated with endogenous condensates, we measured the size of nuclear speckles before and after TSA treatment using a HEK cell line with CRISPR-Cas9-based tagging of eYFP at the SRRM2 gene locus[50] (Fig. 2l). We found that the size of the nuclear speckles significantly decreased after the chromatin network was homogenized by TSA, while the control group without drug treatment showed no change in size(Fig. 2m). In contrast, we increased chromatin network heterogeneity by treating cells with a 4% sorbitol solution, as shown in 13(a) and (b). As expected, increasing chromatin heterogeneity led to an increase in the size of Cajal bodies, endogenous nuclear condensates, as shown in Supplementary Fig. 13c. This result aligns well with the findings from the Corelet condensates (Fig. 2e–h). Taken together, our results suggest that homogenization of the chromatin network impedes condensate growth rates and reduces their sizes.

### Homogenization of chromatin network hinders condensate mobility

Given the observed changes in condensate formation due to alterations in chromatin network heterogeneity, we next examine whether there are also changes to the mobility of condensates. To test this, we light-activate Corelet condensates, wait until no new condensates can be formed, and subsequently capture their motion for 1 min. We find that condensates in TSA-treated cells explore a smaller area compared to untreated cells (Fig. 3a). Additionally, utilizing image-based particle tracking techniques, we analyze condensate dynamics and calculate their MSD. We fit the MSD into power law model for a preliminary comparison between the dynamics of Corelets before and after TSA treatment. Our results show a decreased MSD with TSA treatment (Fig. 3b), which is reflected by statistically significant decreases in both the diffusion coefficient and diffusive exponent (Fig. 3c, d).

To investigate whether the dynamics of engineered condensates resemble that of endogenous condensates, we label Cajal bodies by expressing eYFP-Coilin in cells and track their behavior before and after TSA treatment. We find Cajal bodies are generally excluded from the chromatin network (Supplementary Fig. 14a). This can also be confirmed by the negative correlation between eYFP-Coilin and miRFP670-H2B fluorescence intensity (Supplementary Fig. 14b), consistent with previous findings[69]. Given this exclusionary interaction, resembling that of engineered condensates, we anticipate comparable dynamics. Indeed, we find that Cajal bodies in TSA-treated cells show reduced explored areas compared to untreated cells (Fig. 3e). Additionally, Cajal bodies in TSA-treated cells show a reduced MSD compared to those in untreated cells (Fig. 3f). Correspondingly, there is a decrease in the diffusion coefficient (Fig. 3g), while the diffusive exponent remains unchanged (Fig. 3h). The measured diffusive exponent is consistent with values reported in the literature[74,75]. These findings suggest that both native and engineered condensates experience more restricted motion upon chromatin homogenization with TSA.

To investigate if similar dynamics occur in an assembly attached to the chromatin network, we examined telomeres, which are protected ends of chromosomes that assemble into compact structures of a few hundred nanometers[76–78]. Telomeres were labeled by expressing

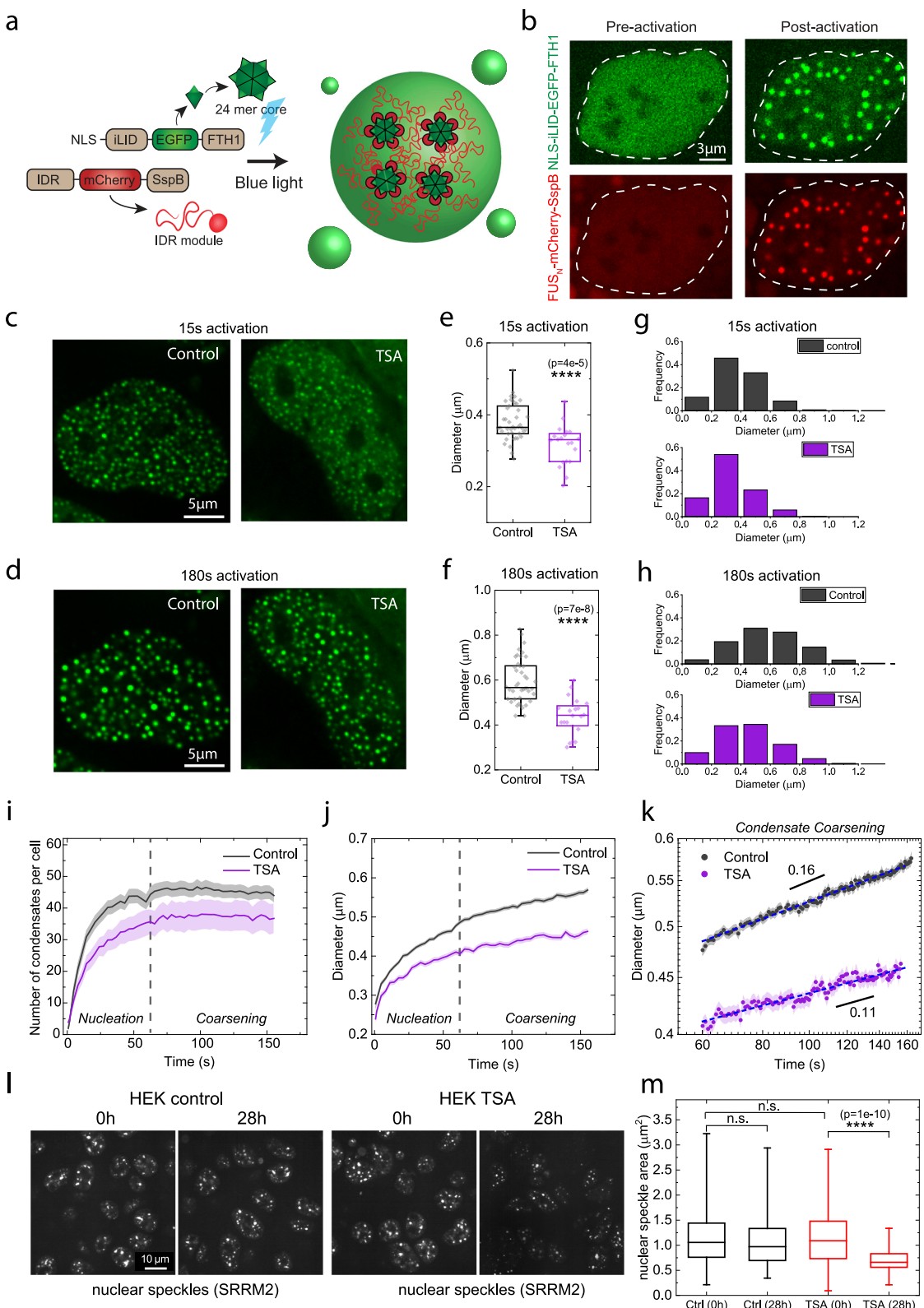

miRFP670-TRF2 in cells, and their dynamics were tracked. We observe that telomeres exhibit a significant reduction in explored area in TSA-treated cells compared to untreated cells (Fig. 3i). In addition, the mean squared displacements (MSDs) of telomeres in TSA-treated cells are smaller than those in untreated cells (Fig. 3j). Both the diffusive exponent and diffusion coefficient are statistically significantly smaller in TSA-treated cells compared to untreated cells (Fig. 3k, l). Overall, these findings suggest that homogenization of the chromatin network

can impede the dynamics of both chromatin-excluding condensates, including engineered and native condensates such as Cajal bodies, as well as chromatin-attaching structures such as telomeres.

## The binodal phase boundary shifts in response to homogenization of the chromatin network

Given these above findings clearly linking condensate growth dynamics and mobility to chromatin compaction state, we ask whether

**Fig. 2 | Impaired condensate growth rates and sizes resulting from homogenized chromatin network. a** Schematics of Corelet system adapted from Bracha et al.[52]. **b** Representative fluorescence images of Corelet expressed in U2OS cells. Representative images of FUS$_N$ Corelets in U2OS cells before and after TSA treatment, with 15 s (**c**) or 180 s (**d**) of light activation. Averaged condensate diameter per U2OS cell before (n = 42) and after TSA treatment (n = 20), with 15 s (**e**) or 180 s (**f**) of light activation. Distribution of condensate diameter in U2OS cells before (n = 42) and after TSA treatment (n = 20), with 15 s (**g**) or 180 s (**h**) of light activation. **i** Number of condensates in U2OS cells under TSA-treated and untreated conditions, with 180 s of light activation. **j** Mean diameter of condensates in U2OS cells before and after TSA treatment, with 180 s of light activation. **k** Mean diameter of condensates during coarsening in U2OS cells before and after TSA treatment. The

data is well-fit by a power law (dashed line). Black solid lines are visual guides. In (**i, j, k**), s.e.m. is represented by the shaded area. n = 19 cells. Representative images (**l**) and mean nuclear speckle area per cell (**m**) of HEK cells with tagging of eYFP-SRRM2 at the endogenous locus via CRISPR/Cas9. Cells are imaged 0 h and 28 h with and without TSA treatment. n = 268, 135, 257, 74 cells from 4 biological replicates for Ctrl (0 h), Ctrl (28 h), TSA (0 h) and TSA (28 h), respectively. In the box plots (**d, g, m**), 25%, 50%, and 75% of the data are indicated using hinges and center lines. The whiskers extend to the highest and lowest values. In (**d, g, m**), p > 0.05 is non-significant. * for p < 0.05, ** for p < 0.01, *** for p < 0.001, and **** for p < 0.0001 based on two-sided student's t-test. For (**b, f, l**), the experiment was repeated three times independently with similar results. For (**d, e, g, h, i, j, k**), three biological replicates are used for statistical analysis.

the phase equilibrium has changed. We take advantage of the ability of the Corelet system to quantitatively map phase boundaries in living cells[56]. We vary the Core-to-IDR ratios of Corelet components and induced phase separation in cells using blue light at different time points post-TSA treatment: 13, 17, 21, 25, and 29 h, which corresponds to various degrees of chromatin network homogenization as we previously measured. We observe that cells continue to undergo phase separation up to 25 h post-TSA treatment; however, phase separation ceases thereafter, as shown by the fluorescence images of cells post blue light activation (Fig. 4a). Interestingly, we observe a consistent decrease in core protein concentration in the dense phase over time, while the concentration in the dilute phase consistently increases (Fig. 4b).

The change of concentration in the dense phase and dilute phase indicates a change in the binodal phase boundary. To quantify this change, we developed an approach for high-throughput phase mapping of thousands of cells, to construct a phase diagram. Briefly, one large area of the sample, representing thousands of cells, is imaged with a low-intensity laser to determine the pre-activation concentration of "Core" and "IDR". The same area is imaged again using a 488 nm laser at 100% intensity for 1 s to induce phase separation of cells. This method allows us to perform a high-throughput survey on thousands of cells in one single large image with 3 channels in 10 min (see "Material and Methods"). Cells are classified as "PS cells" capable of phase separation upon blue light activation, or "nonPS cells" unable to do so, based on whether droplets are detected in individual cells in the confocal images after activation.

Using this high-throughput phase mapping, we can quantitatively map a distinct binodal boundary between the phase separating and non-phase separating populations (Fig. 4c). The boundary is discernible in TSA-treated cells (for 24 h and 36 h) as well (Fig. 4d, e). Comparing phase boundaries before and after TSA treatment, we observe a shift towards higher core concentration with prolonged treatment (Fig. 4f). The minimum core concentration required for phase separation, termed "saturation concentration", increased from 0.10 μM to 0.45 μM with TSA treatment when compared at a fixed core-to-IDR ratio of 1/16 (Fig. 4g); in control cells treated with DMSO for 32 h, we observed no noticeable inhibition of phase separation (Supplementary Fig. 15a, b) and no shift in the phase boundary (Supplementary Fig. 16a, b, c), confirming the presence of DMSO in the TSA solution or potential aging during cell culture did not alter the phase boundary. To confirm that the observed shift in the phase diagram is not limited to FUS-IDR, we switch the IDR of Corelet to HNRNPA1 and determine the phase boundary in cells before and after TSA treatment (Fig. 4h, i). Consistently, the phase boundary exhibited a discernible shift towards higher core concentration, signifying the suppression of phase separation (Fig. 4j). Importantly, we note that the IDR segments we used (FUS, HNRNPA1) are truncated versions of the full-length proteins, lacking the RNA recognition motif (RRM) domain and thus are less likely to be impacted by RNA concentrations in the cell. A similar shift in the phase boundary, indicating the suppression of phase separation, has been observed in HeLa cells treated with TSA,

confirming that this phenomenon is not dependent on the cell line (Supplementary Fig. 17a, b, c, d, e).

To confirm the chromatin network homogenization contributes to the observed change, we also utilized an alternative approach using DZNep, which is known to decompact the chromatin network through a different mechanism. In particular, DZNep inhibits histone methyltransferases (HMTs) and thereby reduces levels of methylated histones H3K27me3[79], resulting in chromatin decompaction[80]. We label the chromatin network with miRFP670-tagged H2B protein in cells and compare it before and after adding DZNep, and observe a decrease in chromatin heterogeneity after treatment (Supplementary Fig. 18a, b). Following this, we again conduct high-throughput phase mapping to determine the phase boundary in cells without and with DZNep treatment (Fig. 4k, l). We observe a noticeable shift in the phase boundary towards higher core concentration following DZNep treatment (Fig. 4m), suggesting inhibition of phase separation. In contrast, we used a 4% sorbitol solution to increase chromatin network heterogeneity, as shown in Supplementary Fig. 19a, b. As expected, we observe that increasing chromatin heterogeneity promotes phase separation, as demonstrated in Supplementary Fig. 19c–e. Taken together, these results provide clear evidence that the homogenization of the chromatin network significantly suppresses the phase separation process, irrespective of the mechanism employed for chromatin decompaction or the specific sequence of intrinsically disordered regions (IDRs) involved in condensate formation or cell line used.

## Loosening the constraint of chromatin at the nuclear boundary does not change the phase diagram

Chromatin is known to be specifically attached to lamin proteins located at the nuclear cortex[75]. We thus sought to explore the impact of reducing these chromatin network constraints on nuclear phase behavior, using siRNA to knock down Lamin A/C proteins. To this end, miRFP670-tagged cells were transfected with 25 nM LMNA siRNA for 3 days. Quantitative analysis reveals a ~ 75% decrease in lamin fluorescence intensity at the nuclear periphery with LMNA siRNA knockdown, while untreated cells or negative control cells treated with non-targeting siRNA were unaffected (Fig. 5a, b). Further increasing the dose of LMNA siRNA to 100 nM does not result in additional knockdown of Lamin A/C (Supplementary Fig. 20a, b, c, d). Interestingly, we quantify chromatin heterogeneity and find no statistically significant difference (Fig. 5c).

These data suggest that lamin proteins primarily affect chromatin organization at the nuclear periphery, rather than the interior. Consistent with this, trajectories of GEM40 nanoparticles showed no difference in the explored area between negative control and Lamin A/C knockdown cells (Fig. 5d), reflecting negligible change in the chromatin microstructure. Similarly, we do not observe difference in the overall diffusive exponent and diffusion coefficient (Fig. 5e, f), indicating the preservation of mesoscale structure in the chromatin network. Additionally, we find that Lamin knockdown has little effect on the phase boundary of Corelet condensates (Fig. 5g–i), likely due to

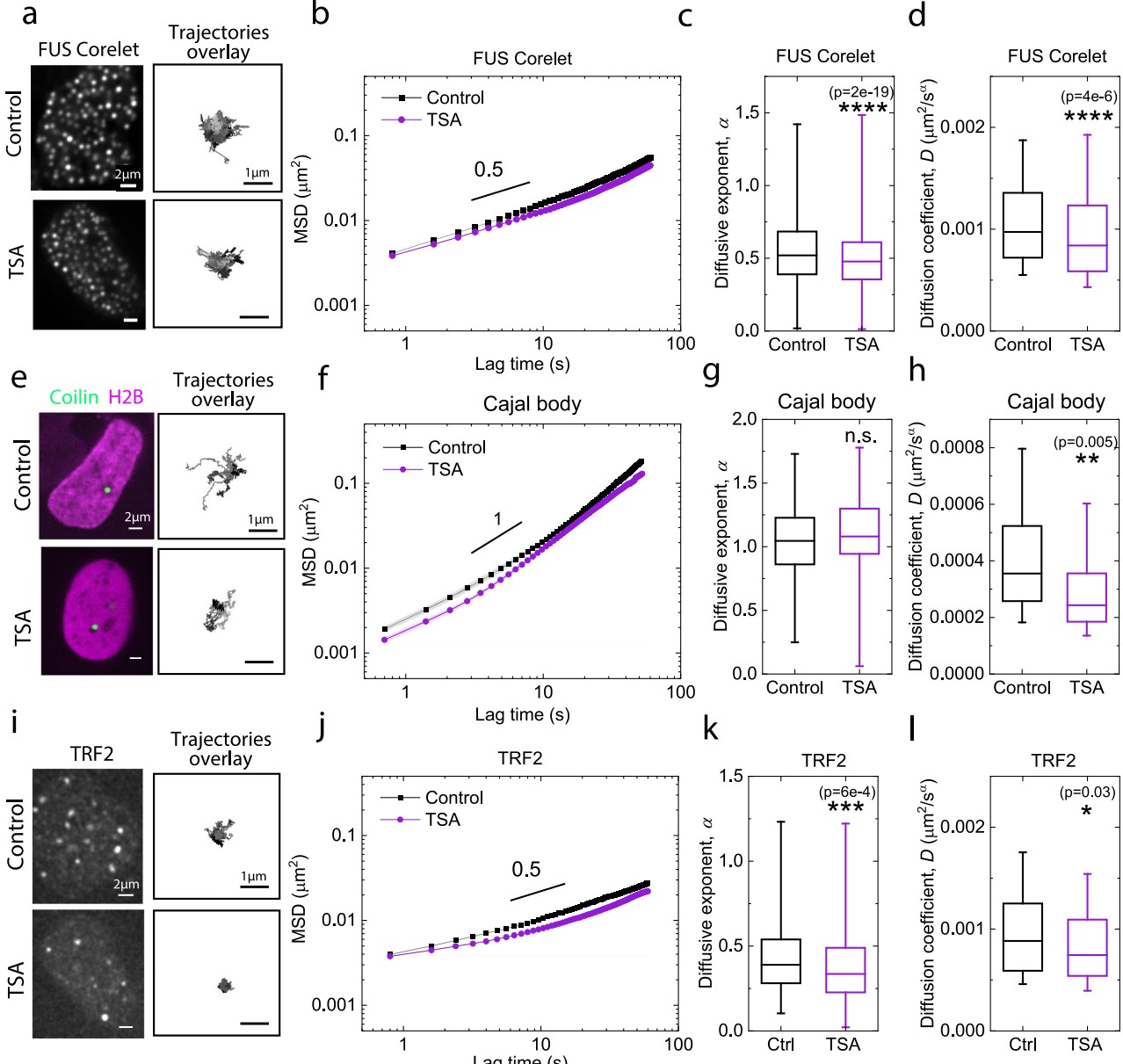

**Fig. 3 | Decreased mobility of condensates and telomeres in U2OS cells due to homogenization of chromatin network. a** Overlay of Corelet trajectories in cells before and after TSA treatment. Mean square displacement (MSD) (**b**), diffusive exponent (**c**) and diffusion coefficient (**d**) of $FUS_N$ Corelets before (n = 20 cells) and after TSA treatment (n = 12 cells), calculated using pairwise MSD. The black solid line is a visual guide. **e** Representative fluorescence images of Cajal bodies before and after TSA treatment. Cajal trajectories overlay are shown to the right. MSD (**f**), diffusive exponent (**g**) and diffusion coefficient (**h**) of Cajal bodies before (n = 132 cells) and after TSA treatment (n = 133 cells) are calculated using single-point MSD. The black solid line is a visual guide. **i** Representative fluorescence images of telomeres before and after TSA treatment. Telomere trajectory overlays are shown to

the right. MSD (**j**), diffusive exponent (**k**) and diffusion coefficient (**l**) of telomeres before (n = 15 cells) and after TSA treatment (n = 25 cells) are calculated using pairwise MSD. Black solid line is a visual guide. In (**b**, **f**, **j**), s.e.m. is represented by the shaded area. In the box plots, 25%, 50%, and 75% of the data are indicated using hinges and center lines. The whiskers extend from the hinges to the highest and lowest values for (**c**, **g**, **k**), and the whiskers extend from the hinges to the 10% and 90% of the data for (**d**, **h**, **l**). In (**c**, **d**, **g**, **h**, **k**, **l**), asterisks indicate two-sided p values based on student's t-tests. p > 0.05 is considered non-significant. * for p < 0.05, ** for p < 0.01, *** for p < 0.001, and **** for p < 0.0001. For (**a**, **e**, **i**), the experiment was repeated three times independently with similar results. For (**b**, **c**, **d**, **f**, **g**, **h**, **j**, **k**, **l**), three biological replicates are used for statistical analysis.

---

minimal changes in chromatin heterogeneity. Taken together, these data suggest that knocking down Lamin A/C protein does not change chromatin heterogeneity and therefore does not affect phase behavior.

### A physical picture of the change in phase behavior upon chromatin decondensation

Our findings provide a physical picture highlighting how changes in the density and mesh size of the chromatin network affect nuclear

phase behavior and condensate dynamics. Condensates tend to form in regions with lower chromatin density and larger mesh size, indicative of softer regions. Homogenizing the chromatin network increases density in these regions, making them stiffer[81], and raising the energy barrier for new condensate formation. Consequently, de novo condensates are smaller, and condensate dynamics are dampened. Further homogenization can inhibit phase separation altogether. These changes reflect a shift of the binodal phase boundary due to the increased free energy of forming condensates locally in mechanically

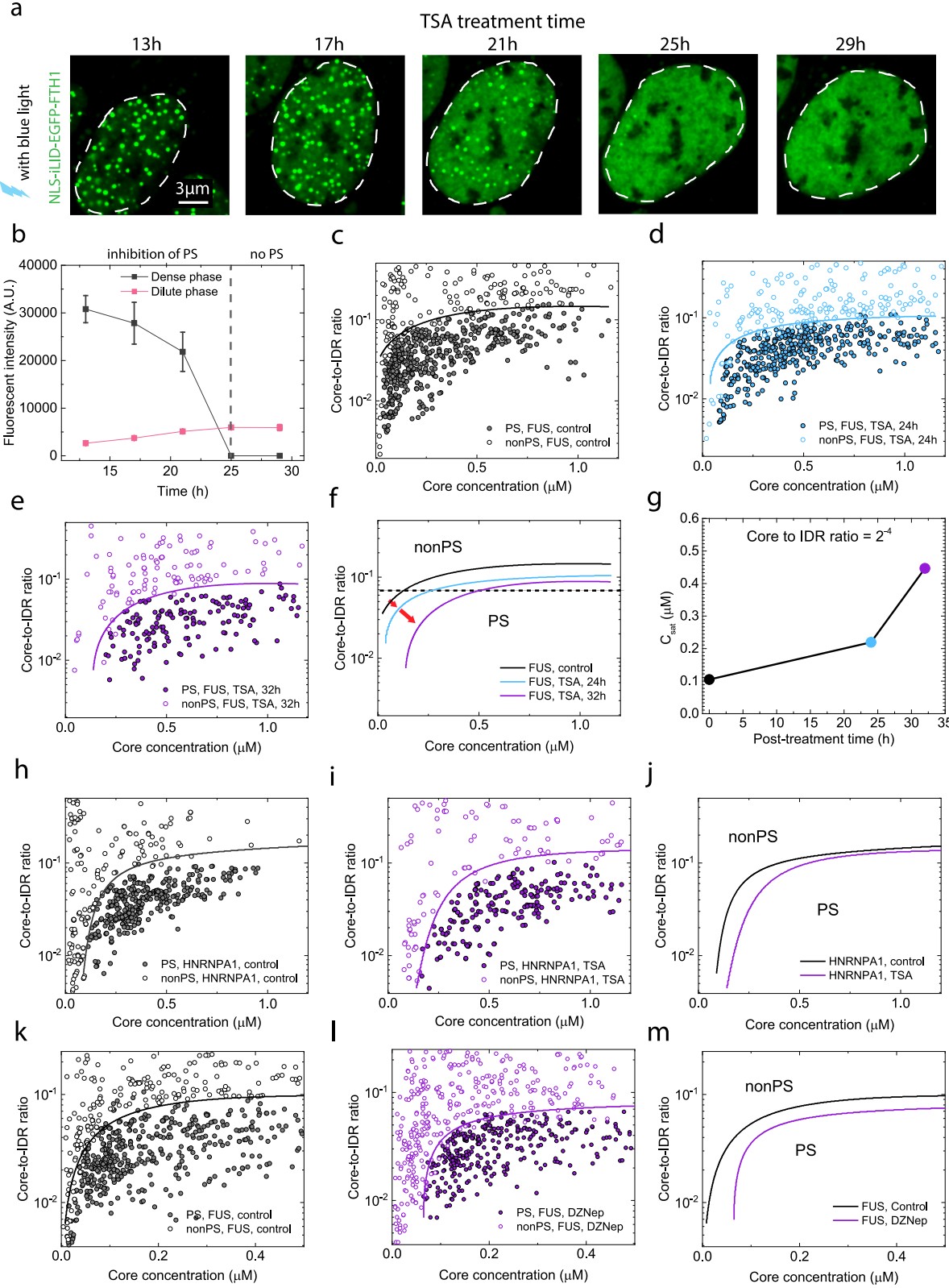

stiffer chromatin (Fig. 6). Overall, our findings demonstrate that the mesoscale structure of the chromatin network has a profound influence on both the phase behavior and dynamics of condensates.

## Discussion

In this study, we employ a range of advanced technologies integrating quantitative live cell imaging, intracellular phase mapping, genetically encoded particle mobility as well as pharmacological and siRNA knockdown perturbations, to investigate the relationship between chromatin heterogeneity and the behavior of biomolecular condensates within the nucleus. Our results reveal that decreasing chromatin network heterogeneity correlates with impeded growth rate of condensates and reduced sizes of de novo condensates. Furthermore, the mobility of condensates is greatly hindered, which is observed in

**Fig. 4 | Inhibition of phase separation and change in phase diagram of U2OS cells due to homogenization of chromatin network. a** Fluorescence images show the formation of light-inducible condensates within the same cell nuclei after blue light activation, treated with TSA for durations of 13 h, 17 h, 21 h, 25 h and 29 h. **b** Quantification of fluorescence intensity of dense and dilute phase of the same nucleus (**a**) after TSA treatment. Data represents Mean ± STD (n = 5 cells from three biological replicates). **c–e** Phase diagrams depict the boundary between phase-separated (PS) cells and non-phase-separated (nonPS) cells expressing FUSN IDR Corelets. These diagrams correspond to cells subjected to TSA treatment for durations of 0 h (**c**) (n = 829), 24 h (**d**) (n = 791), and 32 h (**e**) (n = 295). Data shown represents 3 biological replicates. **f** Shift of binodal phase boundary of light-activated FUSN IDR Corelets in cells following TSA treatment of 0 h (**c**), 24 h (**d**), and 32 h (**e**). **g** Change in the Core concentration at the phase boundary for a constant Core-to-IDR ratio of 1/16, as a function of TSA treatment duration. Data are

quantified from (**f**). **h**, **i** Phase diagrams depict the boundary between phase-separated (PS) cells and non-phase-separated (nonPS) cells expressing HNRNPA1 IDR Corelets. The axes represent Core concentration and Core-to-IDR ratio. These diagrams correspond to cells subjected to TSA treatment for durations of 0 h (**h**) (n = 706), and 32 h (**i**) (n = 651). Data represents 3 biological replicates. **j** Shift of phase diagram of light-activated HNRNPA1 IDR Corelets in cells following TSA treatment of 0 h (**h**), and 32 h (**i**). **k**, **l** Phase diagrams depict the boundary between phase-separated (PS) cells and non-phase-separated (nonPS) cells expressing FUSN IDR Corelets. The axes represent Core concentration and Core-to-IDR ratio. These diagrams correspond to cells subjected to DZNep treatment for durations of 0 h (**k**) (n = 863), and 24 h (**l**) (n = 844). Data represents 3 biological replicates. **m** Shift of phase diagram of light-activated FUSN IDR Corelets in cells subjected to DZNep treatment for durations of 0 h (**k**), and 24 h (**l**).

both engineered and endogenous condensates. Most strikingly, homogenizing the chromatin network correlates with shifts in the binodal phase boundary toward higher concentrations, implying suppressed phase separation. This shift in phase boundary is universally observed, independent of the specific IDRs driving condensation, approaches to reduce chromatin network heterogeneity, or cell lines. Interestingly, we find that knocking down cortical Lamin A/C has no significant impact on the mesoscale structure of the interior chromatin network and phase equilibrium of condensates.

Our findings reveal how the mesoscale structural organization of chromatin relates to the phase behavior and dynamics of embedded biomolecular condensates. In a typical mammalian cell, chromatin is highly heterogeneous, exhibiting a spectrum of compaction states that are categorized into low-density euchromatin, such as those marked by H3K9ac, and high-density heterochromatin, marked by H3K9me[2,3]. By using small molecule drugs that target epigenetic marks to induce global chromatin decondensation, we can modulate this mesoscale spatial heterogeneity of the chromatin network. The HDACi drug TSA increases the mesoscale homogeneity of the chromatin network, likely by decompaction of constitutive heterochromatin domains, such as those located at the nuclear periphery. This is supported by our GEM40 tracking data, which show an increase in the mobility of peripheral GEM40 and a decrease in the mobility of interior GEM40 upon TSA perturbation.

Our findings are consistent with prior work in cells, and with a growing number of studies in non-living soft matter systems linking phase separation to local mechanics. For example, in silicone-based polymer systems and simulations, a mechanically stiffer surrounding matrix shifts the binodal boundary to suppress phase separation[40,41,48,49]. In our experiments, the average phase boundary measured using our Corelet system largely reflects the softest, most phase separation permissive euchromatic microenvironments. Upon homogenizing the entire chromatin network with drugs including TSA and DZNep, decompaction of heterochromatin causes a redistribution of the chromatin polymer into these softer euchromatin areas, locally increasing the mechanical stiffness. This suppresses local phase separation, and shifts the overall phase boundary due to the increase in the free energy required for condensate formation. These observations suggest that both the biological and synthetic polymeric systems may be governed by the same physical mechanism for controlling phase equilibrium[40,41].

Homogenizing chromatin using TSA is associated with suppression of condensate mobility and biomolecular phase separation, which is consistent with effectively increasing local stiffness of the cell nucleus. Interestingly, TSA has been proposed to soften the nucleus in some studies. However, this effect varies across different cell lines and conditions. For instance, recent atomic force microscopy (AFM) measurements have shown that after TSA treatment MDA-MB-231 cells exhibit increased stiffness while MCF-7 cells exhibit decreased stiffness[82]. Moreover, directly measuring the mechanical property of

the nuclear interior is challenging, so most measurements are conducted using exterior probes like AFM or micro-pipettes[14,83]. However, measurements from the interior of biomolecular structures can differ from exterior measurements, depending on the structures the probe interacts with[84–86]. An analogy can be drawn to measuring the mechanical properties of a chicken egg: external measurements may indicate a stiffness that is comparable to a rock, while internal measurements reveal a much softer, viscoelastic environment. Indeed, a study using magnetic tweezers to pull injected magnetic beads inside the nucleus found that the nuclear interior actually becomes stiffer after TSA treatment[87], a result in line with the findings we report here.

Living materials, and the cell nucleus in particular, are soft, composite materials that exhibit significant heterogeneity in organization and properties, with complex internal structures that span length scales ranging from nanometers to tens of microns[1,88]. The cell nucleus should thus be considered as a kind of mesoscale porous material, which cannot be characterized with a single bulk stiffness. Taken together, these observations underscore the often-overlooked perspective that this heterogeneous organization must be considered when characterizing nuclear mechanics and examining the mechanobiological impact on cellular function. In our study, we observe the material impact of nuclear heterogeneity on the phase equilibrium and condensate dynamics, which is primarily rooted in the mesoscale structure of the chromatin network in the interior of the nucleus.

The phase behavior within living cells is an area of intense recent interest, with condensates thought to play roles in many dozens of different physiological and disease processes. Our results potentially reveal a mechanism for physically regulating condensates in cells: by modulating chromatin network heterogeneity, cells may promote or inhibit phase separation at specific gene regions, thereby regulating cellular activities[43]. Moreover, the chromatin network can undergo alterations under various stress conditions, such as chemical stress[89], osmotic stress[90,91], and mechanical stress during cell migration[92]. Concurrently, condensates can also form and undergo changes under these conditions[93,94]. It is worth noting that chromatin homogenization in our assay occurs over a time scale of 24-30 h, which is particularly relevant to the early stages of stem cell differentiation and cellular responses to mechanical cues, as these processes involve similar time scales of chromatin reorganization[19,95]. Our findings suggest possibilities where the chromatin network could be harnessed as a modifiable component for cells to regulate the behavior of condensates. This may bridge the gap between cellular stress, mechanics, and the regulation of biomolecular condensates. Additionally, this research provides valuable insights for materials science studies. It highlights how the structure and mechanical properties of porous media can influence liquid-liquid phase separation and the dynamics of resulting condensates. This opens up avenues for designing and engineering biomaterials and bio-inspired materials that guide condensate formation. These insights not only shed light on the intricate connection between the material state of biomaterials and phase separation, but also offer

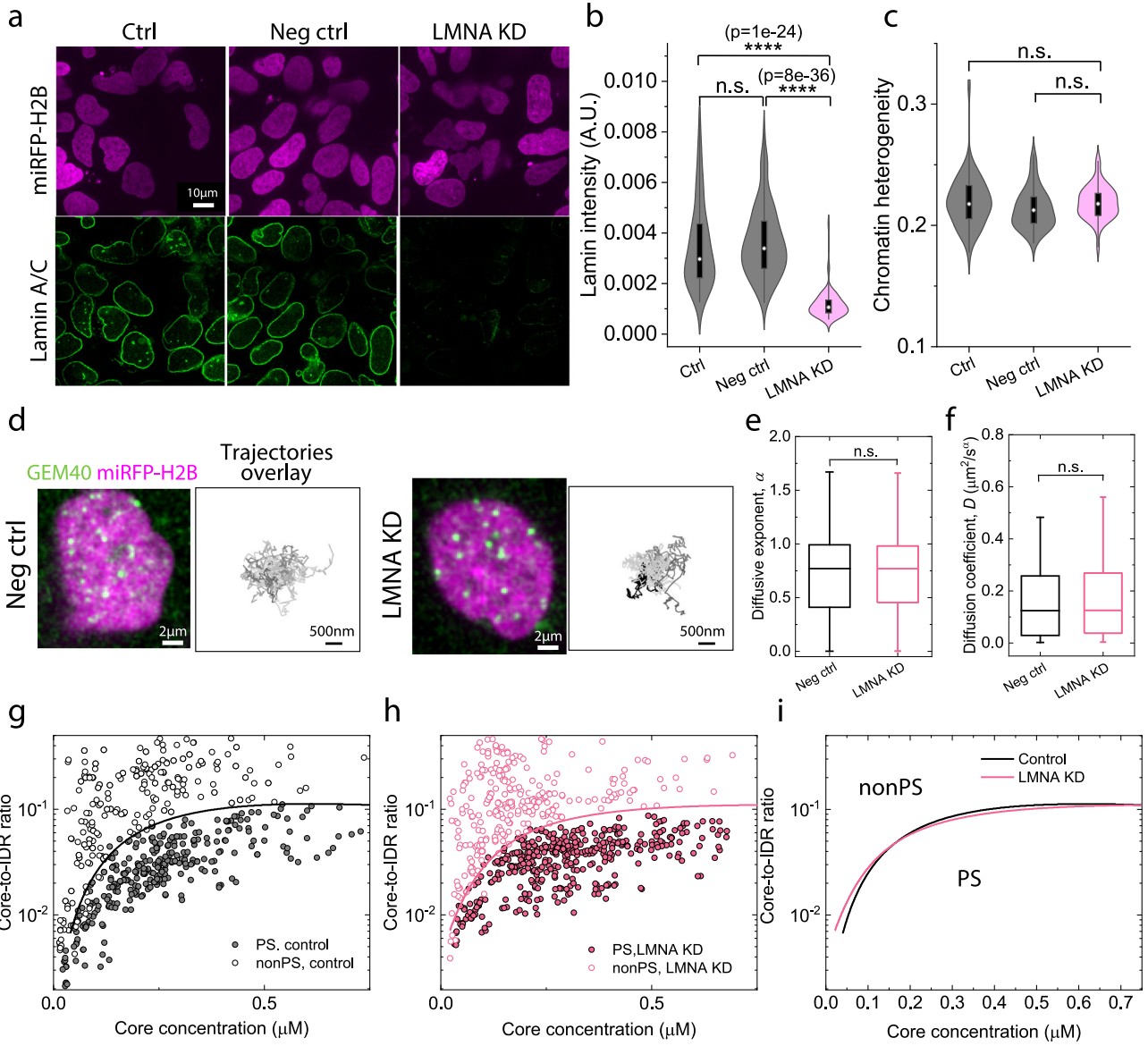

**Fig. 5 | Loosening the constraint of chromatin at the nuclear boundary does not change the phase diagram of U2OS cells. a** Fluorescence images showing the immunofluorescence staining of Lamin A/C in untreated (Ctrl), non-target siRNA (Neg Ctrl) and Lamin A/C knockdown (LMNA KD) cells. **b** Lamin A/C fluorescence intensity per cell in the nuclear periphery of untreated control (n = 67), negative control (25 nM non-targeting siRNA, n = 99), and Lamin A/C knockdown cells (25 nM LMNA siRNA, n = 86). **c** Nuclear heterogeneity of untreated control (n = 51), negative control (n = 49), and Lamin A/C knockdown cells (n = 46). **d** Representative fluorescence image of GEM40 in miRFP670-H2B-expressing cells. Negative control and Lamin A/C knockdown cells are shown. Additionally, the overlay of GEM40 trajectories in each cell is presented. **e, f** Diffusive exponent and diffusion coefficient of GEM40 in negative control (**e**) (n = 28) and Lamin A/C knockdown cells (**f**) (n = 54). **g, h** Phase diagrams depicting the boundary between phase-

separated (PS) cells and non-phase-separated (nonPS) cells transfected with $FUS_N$ Corelets. The axes represent Core concentration and Core-to-IDR ratio. These diagrams correspond to wild-type cells (**g**) and cells with Lamin A/C knockdown (**h**). **i** Comparison of phase boundary of $FUS_N$ Corelets in wild-type cells (**g**) (n = 656) and cells with Lamin A/C knockdown (**h**) (n = 847). In the box plots, 25%, 50%, and 75% of the data are indicated using hinges and center lines. The whiskers extend from the hinges to the highest and lowest values for (**b, c, e**), and the whiskers extend from the hinges to the 10% and 90% of the data for (**f**). In (**b, c, e, f**), p > 0.05 is non-significant, * for p < 0.05, ** for p < 0.01, *** for p < 0.001, and **** for p < 0.0001. For (**a, d**), the experiment was repeated three times independently with similar results. For (**b, c, e, f, g, h**), three biological replicates are used for statistical analysis.

potential implications for therapeutic research and applications targeting cellular condensates.

## Methods

### Plasmids

pHR-NLS-iLID-mGFP-FTH1, pHR-FUSN-mCherry-SspB, and pHR-HNRNPA1C-mCherry-Sspb are generated in our previous studies[52]. FM5-H2B-miRFP670 and FM5-TRF2-miRFP670 are kind gifts from Yoonji Kim. FM5-GEM40-NLS-mGFP is a kind gift from Amal

Narayanan. FM5-Coilin-eYFP is a kind gift from Lennard Wiesner. FUCCI is a kind gift from Jessica Z. Zhao.

### Cell culture and drugs

Lenti-X™ 293T cell line (Takara Bio USA) is used to produce lentivirus. U2OS (ATCC, VA, USA) cells and HeLa (ATCC, VA, USA) cells are used for experiments. All cell lines were cultured in complete growth media, comprising Dulbecco's Modified Eagle Medium (DMEM, Thermo Fisher Scientific Inc, MA, USA) supplemented with 10% fetal bovine

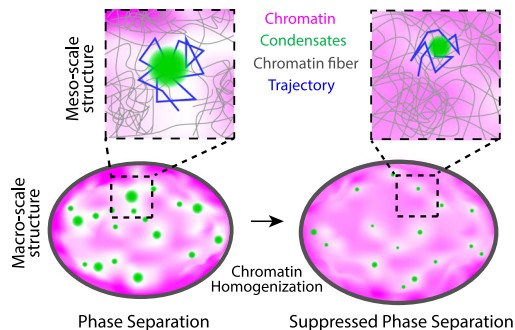
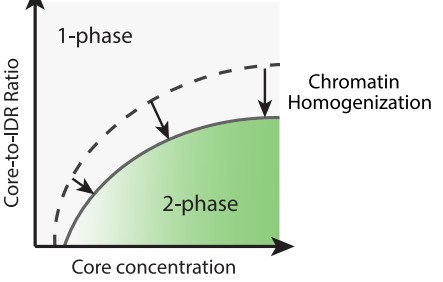

**Fig. 6 | A physical picture depicting the change in phase behavior upon homogenization of chromatin network.** Schematic diagram illustrating the meso-scale and macro-scale change of the chromatin network due to homogenization. Homogenization of the chromatin network results in a decrease in the size and mobility of condensates, and eventually leads to the inhibition of phase separation, manifested in a shift of the binodal phase boundary to the right.

serum (FBS, Atlanta Biologicals, GA, USA) and 1% penicillin-streptomycin (Thermo Fisher Scientific Inc, MA, USA). The cells were maintained at 37 °C with 5% CO2 in a humidified incubator. Trichostain A (TSA) ready-made solution (5 mM, Sigma-Aldrich, MO, USA) was diluted 5000-fold in complete growth media. The diluted solution was then added to the cells during the experiment. 3-Deazaneplanocin A (DZNep, Cayman Chemical, MI, USA) was prepared by diluting it to a concentration of 5 μM using complete growth media sourced from a 10 mM stock solution. HEK293 cells with endogenously eYFP-tagged SRRM2 were generated using CRISPR-Cas9, following the method previously established in our lab[1].

### Lentivirus production and cell transfection
To produce lentivirus, Lenti-X™ 293T cells are initially seeded onto a 6-well plate and cultured until reaching 60% confluency. Plasmids, including 1.72 μg of the target construct, 500 ng of pRSV-REV, 1 μg of pMDLg/pRRE, and 600 ng of pMD2.G (VSV-G), are combined with 172 μL of Opti-MEM (Thermo Fisher Scientific Inc, MA, USA). Additionally, 6.9 μL of P3000 (Invitrogen, CA, USA) is added to the mixture, which is then thoroughly mixed to create a DNA mix in a test tube according to the manufacturer's protocol.

In another test tube, 6.9 μL of Lipofectamine 3000 (Invitrogen, CA, USA) is combined with 172 μL of Opti-MEM. The DNA mix is then added to the Lipofectamine 3000 mixture and incubated for 15 min. Subsequently, the final mix is added to the Lenti-X™ cells in a dropwise manner. The cells are cultured for an additional 2 days to allow for lentivirus production.

Following the incubation period, the cell culture media containing the lentivirus is collected and filtered through a 0.45 μm pore size filter to remove cellular debris. The filtered media is then concentrated to 4X using the Lenti-X™ Concentrator (Takara Bio USA) following the manufacturer's protocol, for immediate usage or stored at −80 °C.

### Calibration of protein concentration
The mapping of fluorescence intensity to absolute concentration was conducted using a U2OS cell line expressing mGFP-P2A-mCherry, featuring an equimolar amount of intracellular mGFP to mCherry owing to the autocatalytic P2A linker. Measurements were carried out using Zeiss LSM 980 laser-scanning confocal microscopes (Carl Zeiss Microscopy, LLC, NY, USA), equipped with a C-Apochromat 40×/1.2 W autocorr FCS M27 objective. All measurements and data analysis were performed using Zeiss ZenBlue Dynamic Profiler Software with an AiryScan detector.

To measure the fluorescent protein concentration, a reference image of the cells was captured at 488 nm, with regions of interest selected within the cell nuclei for subsequent measurement. Subsequently, protein concentration data were acquired using a 10-s FCS measurement time. Intracellular mGFP concentrations (in nM) were determined by fitting the autocorrelation curve to a 1-component 3D diffusion model using the Dynamic Profiler (Carl Zeiss Microscopy, LLC, NY, USA) software. The intracellular concentration of mCherry was derived by determining the mCherry to GFP fluorescence ratio.

Further fluorescence calibration between the Zeiss LSM 980 laser-scanning confocal microscopes and the spinning disk confocal microscope (utilized for Corelet phase mapping) was performed through linear fitting of the fluorescence intensities of the mGFP-P2A-mCherry U2OS cells captured on both microscopes.

### Western blot and quantification
To quantify the expression levels of H2B and miRFP670-H2B in cells, we performed Western blot analysis against H2B using three independent replicates. Cell pellets from each group were resuspended and incubated in RIPA buffer (Thermo Scientific #89901) containing a 100× Halt™ protease and phosphatase inhibitor cocktail (Thermo Scientific #78440) and 300× benzonase nuclease (Millipore Sigma #E8263) for 30 min on ice. Following BCA protein quantification (Pierce), 10 μg samples were denatured using LDS sample reducing agent (Invitrogen #NP0004) and sample buffer (Invitrogen #NP0007). For H2B detection, an anti-H2B primary antibody (Cell Signaling Technology #12364) was used at a dilution of 1:2000. For miRFP detection, a miRFP antibody (Thermo Fisher Scientific Inc, PA5-109200) was used at a dilution of 1:1000. The secondary antibody (Jackson #111-035-144) was diluted 1:10000. As a loading control, β-Tubulin was stained using a β-Tubulin antibody (Cell Signaling Technology #2146) at a dilution of 1:1000, with the same secondary antibody used at 1:10000.

### Cell number quantification and proliferation assay
Cell numbers were quantified using the commercial CCK-8 kit (Dojindo Laboratories) according to the manufacturer's instructions. Briefly, cells were seeded into multiple 96-well culture plates with 200 μL of media per well and incubated at 37 °C in a humidified incubator with 5% CO2 for 24 h to allow for adherence. On different days of continuous culture, 10 μL of CCK-8 solution was added to each well of a plate, which was then incubated for 3 h before measurement. The absorbance of each well was measured at 450 nm using a microplate reader (Agilent BioTek Synergy H1), with absorbance values being directly proportional to the number of living cells. The plate was discarded after measurement and not reused. In a separate plate, cells were pre-seeded at known numbers (100,000; 50,000; 25,000; 12,500; 6250) per well, and the absorbance was measured for each well. A calibration curve was subsequently generated based on the measured absorbance and known cell numbers in each well. This

calibration curve was used to convert the absorbance of the experimental samples to cell numbers.

## Cell viability assay

To measure cell viability, the Live/Dead Assay Kit (Abcam) was used according to the manufacturer's instructions. The Live and Dead Dye was diluted 1:200 in prewarmed cell culture media, and the cells were incubated with the dye for 10 min. The stained samples were then imaged using a confocal microscope (see "Microscope setup"). Live cells were imaged in the 488 nm channel, while dead cells were imaged using the 561 nm channel. Images were analyzed using ImageJ to calculate the number of live and dead cells in the sample.

## Cell cycle assay

To quantify the cell cycle, we utilized the Fluorescent Ubiquitination-based Cell Cycle Indicator (FUCCI) system[2]. Cells were transfected with FUCCI plasmids via lentiviral delivery, enabling visualization of cell cycle phases through fluorescence. The FUCCI system uses two fluorescent proteins: one labels cells in the G1 phase, and the other labels cells in the S/G2/M phases. Cells were seeded into 96-well plates and allowed to adhere and spread. Fluorescence microscopy was employed to visualize and capture images of the cells, distinguishing between different cell cycle phases based on their emitted fluorescence colors. Quantification of the cell cycle distribution was performed using ImageJ.

## Construction of stable cell lines

To transfect the cells with lentivirus for experiments, U2OS cells are initially plated on a glass-bottom 96-well plate at 30% confluency. Subsequently, 25 μL of the media containing concentrated lentivirus is added to each well. The cells are then cultured for an additional 2 days, allowing time for virus transduction. Following the incubation period, the virus-containing medium is replaced with fresh growth media. Cells infected were typically imaged no earlier than 72 h after infection.

## Sorbitol assay to increase chromatin heterogeneity

Sorbitol powder was dissolved into fresh, complete cell culture media at 4% w/w. Cells are seeded onto a 96-well plate and loaded onto the microscope. Sorbitol solution was added to cells and the sample is imaged after a 30-min incubation period to allow for thermal equilibrium. Cells were subsequently imaged using the high-throughput phase mapping protocol before and after adding sorbitol.

## Imaging and image analysis

**Microscope setup.** Cells were imaged using a custom-built spinning disk confocal microscope comprising a Nikon Plan Apo VC 100×/1.4 oil immersion objective (Nikon Instruments Inc., NY, USA), a Yokogawa CSU-W1 Confocal Scanner Unit (Yokogawa, PA, USA), and an Andor DU-897 electron-multiplying charge-coupled device camera (Oxford Instruments, MA, USA) mounted on a Nikon Eclipse Ti body (Nikon Instruments Inc., NY, USA).

For live-cell imaging experiments, an Okolab cage incubator was integrated on top of the microscope to maintain a controlled environment of 37 °C and 5% CO2, ensuring optimal conditions for cell viability and function.

**Fluorescence recovery after photobleaching (FRAP).** The FRAP assay was conducted using a Zeiss LSM 980 laser-scanning confocal microscope. Cells expressing miRFP670-H2B were seeded in a 96-well plate. The cells were stained with Hoechst 33342 (Thermo Fisher Scientific Inc) in culture media for 15 min and then washed with fresh media three times (at intervals of 5 min, 30 min, and 1 h) to remove unbound Hoechst 33342. Bleaching was performed on circular regions of interest (ROIs) with a 1 μm radius using the 405 nm and 639 nm lasers at 100% power for 5 frames, with a reference ROI in the same cell

that was not bleached. Fluorescence recovery was monitored by imaging the Hoechst and miRFP channels for 60 s. Fluorescence intensity was background-subtracted and normalized based on the lowest and highest fluorescence intensity of the ROI. Each normalized curve was fitted with the equation: $y = b*(1-e^{-k(t-t0)})$ to calculate the mobile fraction of the bleached molecule, where $b$ is the mobile fraction and $t0$ is the starting frame of bleaching.

**GEM40.** For the GEM40 dynamics experiment, a snapshot of chromatin and GEM40 was acquired using a 488 nm laser at 50% intensity, with an exposure time of 200 ms, and a 640 nm laser at 30% intensity, with an exposure time of 200 ms. To capture the rapid dynamics of the GEM40 particles, cells were imaged using the fast timelapse feature of the NIS-Elements software (Nikon Instruments Inc., NY, USA), employing a 488 nm laser at an intensity of 50% with an exposure time of 20 ms.

To extract the trajectories of the GEM40 particles, the time-lapse images were Gaussian blurred to reduce noise and analyzed using the Particle Tracker plugin in ImageJ (version 1.53 h, https://imagej-nih-gov.ezproxy.princeton.edu/ij). The extracted trajectories were further analyzed using custom-written code in MATLAB to calculate the mean square displacement *MSD* and anomalous diffusive exponent $\alpha$ and diffusion coefficient $D$, where MSD $\leq x^2 \geq \propto Dt^{\alpha}$.

To characterize the spatial heterogeneity of GEM40 dynamics, images of nuclei marked by miRFP670-H2B were Gaussian blurred with a 1-pixel kernel and segmented using the Otsu's method in ImageJ to generate the nuclear mask. The segmented nucleus was then eroded with a 3-pixel (~0.4 μm) kernel to generate the mask of the nuclear interior. The mask of the nuclear periphery was generated by subtracting the nuclear interior mask from the nuclear mask. GEM40 particles were classified into two groups based on their spatial position relative to the nuclear interior and periphery masks. Statistical analysis was subsequently performed for each group of classified GEM40 particles.

**Corelet condensate size, number, and dynamics.** Samples are prepared by transducing Corelet components plasmids into U2OS cells using lentivirus (see "Construction of stable cell lines"). To visualize the growth of condensates in cells, time-lapse movies are captured at a frame rate of 1.2 fps. A 488 nm laser at 25% intensity, with an exposure time of 200 ms, is used for most assays. However, in assays where chromatin needs to be imaged concurrently with condensate growth, a 640 nm laser at 30% intensity, with an exposure time of 200 ms, is utilized as well and alternated with the 488 nm laser per frame. The cells are imaged for a total duration of 3 min to record the time-dependent growth of condensates.

To quantify the size and number of condensates, individual nuclei are manually cropped from the time-lapse movies and saved as.tif files. Subsequent analysis is performed using ImageJ. The cropped images are Gaussian blurred with a 1-pixel kernel to reduce noise. Each cropped time-lapse stack is segmented using a fixed threshold determined at 1 min after activation (when the condensate size reaches quasi-equilibrium) using the Renyi method. The number and size of condensates per frame in each cell are then analyzed using the particle analyzer module in ImageJ.

To analyze the dynamics of the condensates, the first 1-min movie is discarded, and only the last 2-min movie is analyzed. Individual nuclei are cropped manually, Gaussian blurred with a 1-pixel kernel, and thresholded as before using ImageJ. The positions of the condensates in each frame are determined using the cntrd.m function in MATLAB (MathWorks, MA, USA). Custom-written code is then used to calculate the pair-wise mean square displacement (MSD) in MATLAB.

**High throughput phase mapping.** Cells are initially imaged using a 488 nm laser at an intensity of 3% with an exposure time of 200 ms,

along with a 561 nm laser at an intensity of 30% with an exposure time of 200 ms. This imaging setting is performed to determine the fluorescence intensity of the "Core" and "IDR" of individual cells before activating the phase separation process with blue light (pre-activation). The fluorescence intensity data are subsequently converted into physical concentrations using a calibration curve(see Calibration of protein concentration). Following pre-activation imaging, cells are imaged again using a 488 nm laser at an intensity of 100% with an exposure time of 1 s to capture post-activation snapshots. These images are utilized to classify the cells into phase-separated (PS) and non-phase-separated (non-PS) groups. The Core concentration and Core-to-IDR ratio are determined from the corresponding pre-activation snapshots, which are then converted into physical concentrations.

To generate the phase diagram, cells are mapped to a 2-dimensional space based on Core concentration and Core-to-IDR ratio and classified into PS and non-PS groups. To determine the phase boundary in an unbiased manner, the support vector machine (SVM) classifier function (fitcsvm.m) in MATLAB is employed to calculate the boundary that separates the two groups.

**Cajal body dynamics.** Samples are prepared by transducing FM5-Coilin-eYFP and FM5-H2B-miRFP670 plasmids into U2OS cells using lentivirus (see "Construction of stable cell lines"). To visualize the dynamics within cells, time-lapse movies are captured at a frame rate of 1.2 fps. Imaging is performed using a 488 nm laser at 25% intensity, with an exposure time of 200 ms, and a 640 nm laser at 30% intensity, with an exposure time of 200 ms. The cells are imaged for a total duration of 3 min.

For the analysis of Cajal body dynamics, individual nuclei are first identified using the miRFP channel of the image, and then manually cropped. The Cajal bodies marked by Coilin-eYFP in the 488 nm channel are Gaussian filtered with a 1-pixel kernel and tracked using the particle analyzer module in ImageJ to export the trajectories. The exported trajectories are subsequently analyzed using custom-written code to calculate the single-point mean square displacement (MSD) in MATLAB.

**Nuclear volume measurement.** U2OS cells were transduced with lentivirus containing FM5-H2B-miRFP670 plasmids as described in (Methods section "Lentivirus production and cell transfection"). To measure nuclear volume, optical cross-sections were recorded at 0.3 μm intervals along the z-axis using the 640 nm channel to construct a z-stack of each cell. Each nucleus was subsequently cropped and analyzed using ImageJ and MATLAB (MathWorks, MA, USA) with custom-written code to quantify nuclear volume.

**Lamin A/C and chromatin intensity analysis.** To visualize chromatin in cell nuclei, U2OS cells are transduced with lentivirus containing the FM5-H2B-miRFP670 plasmids (see "Construction of stable cell lines"). For imaging Lamin A/C in cell nuclei, cells are fixed and stained for anti-lamin A/C antibody and visualized using a 488-nm secondary antibody (see Immunofluorescence staining).

To capture the 3D distribution of lamin A/C and chromatin, z-stack images with a z-interval of 0.3 μm are acquired. Imaging is performed using a 488 nm laser at 25% intensity, with an exposure time of 200 ms, and a 640 nm laser at 30% intensity, with an exposure time of 200 ms. To generate a video of 3D rotating nuclei or lamin, the images are rendered in 3D, and a 360-degree rotating action is added to generate a.avi format video using Imaris software (Oxford Instruments, MA, USA).

To analyze chromatin heterogeneity before and after TSA treatment, either the middle plane (2D) or the full z-stack(3D) of the cell nucleus is selected for analysis. Individual cells are manually tracked and cropped, and the resulting images are saved as.tif files. These images are then processed using ImageJ. The individual nucleus is Gaussian filtered with a 1-pixel kernel and segmented using the Otsu's thresholding method to generate a mask. The mask, along with the Analyze Particles module (with a minimum size threshold of 10 μm²), is used to measure the standard deviation and mean intensity of the fluorescence signal before segmentation. The intensity of the background is determined from areas with no cells in the image. Chromatin heterogeneity is calculated as the standard deviation divided by the mean of the intensity, with background intensity subtracted.

**Lamin A/C knockdown with siRNA**
Samples are prepared by seeding cells onto a glass-bottom 96-well plate at 60% confluency and allowed to adhere overnight in a cell culture incubator. The siRNA solution, either ON-TARGETplus Human LMNA siRNA (Cat. J-004978-05-0005, Horizon Discovery Ltd, PA, U.S.A.) or ON-TARGETplus Non-targeting Control Pool (Cat. D-001810-10-05, Horizon Discovery Ltd, PA, U.S.A.), is diluted in Opti-MEM and incubated for 5 min at room temperature. Simultaneously, the DharmaFECT reagent (Cat. T-2001-02, Horizon Discovery Ltd, PA, U.S.A.) is diluted in Opti-MEM and incubated for 5 min at room temperature.

Subsequently, the two solutions are mixed and further diluted 5X in an antibiotic-free complete medium, reaching a final volume of 200 μL. The final concentration of the siRNA is 25 nM or 100 nM according to different experiment assays. The original culture medium in the 96-well plate is then removed, and the mixed siRNA solution is added to the cells. The cells are then incubated in the incubator for an additional 48 h before imaging.

**Immunofluorescence staining**
For immunostaining of Lamin A/C, cells are first fixed for 10 min with 4% formaldehyde and 0.1% Triton X-100 diluted in PBS at RT. Subsequently, the cells are washed with PBS three times to remove excessive reagents. After fixation, cells are blocked with 10% normal goat serum (Thermo Fisher Scientific Inc, MA, USA) in PBS for 1 h. Following blocking, a two-step immunostaining process is performed. Fixed cells are incubated with Lamin A/C mouse antibody (4777S, Cell Signaling Technology, MA, USA), diluted 1:200 in 10% normal goat serum, for 1 h at room temperature. Samples are then washed five times for 5 min each with PBS.

Next, the cells are incubated with Goat anti-Mouse Alexa Fluor™ 405 secondary antibodies (Thermo Fisher Scientific Inc, MA, USA), diluted 1:200 in 10% normal goat serum, for 1 h in the dark. The stained cells are washed three times with PBS and imaged using a Nikon spinning disk microscope.

**Statistical analysis and reproducibility**
All experiments were performed in N ≥ 3 independent biological replicates. Statistical analysis is conducted using Origin software(OriginLab Corporation, MA, USA). Student's t-tests are employed for comparing between two groups. P-values greater than 0.05 are considered non-significant in all analyses. P-values less than 0.05 are denoted with *, while those less than 0.01 are denoted with **, those less than 0.001 with ***, and those less than 0.0001 with ****.

**Reporting summary**
Further information on research design is available in the Nature Portfolio Reporting Summary linked to this article.

## Data availability
We have included the source data used for this study in the manuscript. Source data are provided with this paper.

## Code availability
Codes for image analysis and quantification with this work are publicly available via GitHub at https://github.com/xiajing09/Chromatin-heterogeneity-and-phase-behavior.

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

## Acknowledgements

We thank Evangelos Gatzogiannis for his valuable assistance with microscopy. And we thank Amal Narayanan, Yoonji Kim, Mack Walls, David W. Sanders and Lennard Wiesner for their generous help with plasmid preparation. We acknowledge useful discussions from Cornelis Storm (Eindhoven University of Technology) and Chang-hyun Choi, Troy J Comi, Hongboi Zhao, Anita Đonlić and members of the Brangwynne lab. This work was funded by the Princeton Center for Complex Materials, an NSF MRSEC (DMR-2011750); the AFOSR MURI (FA9550-20-1-0241); the St. Jude Research Collaborative on the Biology and Biophysics of RNP granules; and the Howard Hughes Medical Institute.

## Author contributions

J.X. and C.P.B. conceptualized the study. J.X. and J.Z.Z. performed experiments. J.X. performed formal analyses. J.X., J.Z.Z., A.R.S., and C.P.B. wrote the manuscript. C.P.B. acquired the funding for this work.

## Competing interests

C.P.B. is a founder of and consultant for Nereid Therapeutics. J.X., J.Z.Z., and A.R.S. declare no competing interests.
