## [Peer Review file · Nature Communications]

Chromatin Heterogeneity Modulates Nuclear Condensate Dynamics and Phase Behavior

Corresponding Author: Professor Clifford Brangwynne

Version 0:

Reviewer comments:

Reviewer #1

(Remarks to the Author)

In this work, Xia and collaborators investigated the contribution of chromatin heterogeneity to the dynamic formation of chromatin condensates and phase separation shift. Specifically, they considered the potential relevance of the local mechanical properties of chromatin as a modulator for condensates dynamics through phase separation. They further extend their results proposing that the modulation of chromatin heterogeneity may represent a regulatory mechanism to control phase transition, thus impacting on condensates formation.

Although the forwarded biological question is of high relevance as it proposed a central role for the local mechanical forces in regulating nuclear condensates, the experimental design and the provided data are not adequate to properly address this question. The results obtained do not support the raised conclusions, and the authors need to perform a series of controls to strengthen the robustness of the results and the significance of their data. In addition, the work contained multiple approaches and experimental settings which are poorly controlled, which reduced the interest and the relevance of the findings.

In sum, although the proposed role of the local chromatin mechanics in tuning phase separation of nuclear condensates is of high relevance, further experiments are required to improve the soundness of the raised conclusions. Indeed, there are important technical limitations and a lack of appropriate controls that reduce the impact and the robustness of this study that need to be addressed before considering the manuscript ready for publication.

Major criticisms:

1. The authors used a poorly controlled drug-based approaches to impact the chromatin states, namely the usage of the HDAC inhibitor TSA. It is highly documented how this perturbation can affect many aspects of chromatin functions and cellular responses, yet the author did not perform any control to determine the cellular, transcriptional or chromatin response, apart detecting the intensity of the miRFP670-H2B and retrieving an indirect measurement of chromatin heterogeneity. They used a single cell line, in which they transiently overexpressed all the biosensor used in the manuscript, never showing the level of heterogeneity nor the expression level. For example, what is the level of the overexpressed H2B, with respect to the endogenous protein? Is it miRFP670-H2B organized into nucleosomes incorporated into the chromatin? What are the consequences of its over-expression? Having stable clones expressing a quasi-physiological level of the reported is mandatory to reach any conclusion of this work. To better control the system, a knock-in of the miRFP670- into one of the endogenous loci of H2B will ensure the correct measurement of chromatin distribution, in response to any perturbation. Without these controls in at least two different cell lines no reliable conclusion can be raised by this study.
2. The usage of TSA or DZNep inhibitors per se does not implies an "homogenization" of chromatin as stated by the authors. These compounds affect different kinds of chromatin compartments, with TSA causing a general uncontrolled increase of histone acetylation, effecting both eu- and constitutive (H3K9me2/3) heterochromatin. DZNep instead is a more specific inhibitor of EZH2, which deposit H3K27me1, me2 and me3, the latter being involved in Polycomb functioning in maintaining a repressive chromatin environment. To simplify the consequences of the usage of these compounds to a poorly defined and characterized "homogenization" of chromatin is at the best superficial and inadequate. The authors need to characterize the alterations of the chromatin states by imaging and sequencing, after having ensured that the toxicity of the treatment did not alter the cellular responses (vitality, cell cycle progression, cell adhesion ect.). They further should clarify the reasons why they chose these drugs and the rationale behind comparing such a diverse chromatin pattern in the investigated system.
3. The authors interpreted the obtained results, proposing that the uncontrolled perturbation of the chromatin states results in a relevant change of the local mechanical forces that. Impinges on the phase separation and the assembly of condensates.

Unfortunately, they did not provide any direct or indirect measurements of chromatin mechanics in any of the artificial biological contexts.

4. All the biophysical measurements appear to lack the appropriate controls, ensuring the robustness of the raised conclusions. For example, although the high variability associated with transient expression of the used biosensor, the authors did not provide biological replicates. Moreover, they never considered the possible perturbation of the cell cycle state, nor considered that cells in different stages of the cell cycle are characterized by important difference in terms of 3D chromatin organization, histone modifications, and chromatin compaction. How do the authors interpret their data considering that on average a small fraction of the nuclear volume is occupied by the nucleosome in interphase?

5. All the imaging measurements have been performed in 2D, without taking into consideration the spatial (3D) genome organization, especially of heterochromatin, which is characterized by long-range interactions clustering together non-linear portion of the genome. The Authors should perform the imaging-based measurements in 3D.

6. The proposed models are based on a experimental design that investigate the behavior of heterogenous, exogenous proteins. Although the designed tools permit to overcome some intrinsic limitation in measuring the biophysical properties of condensates within the cellular context, the authors did not provide and substantial data demonstrating that the measured behavior represent a general trend of the endogenous counterpart. The inclusion of a reporter of Cajal body or for the telomere based of the over-expression of a reporter system does not entail the authors to extend their conclusion to the endogenous condensates.

7. The perturbation of the chromatin states with these compounds will certainly also impact the gene expression state and transcripts abundances. What is the impact of altering the expression rates on the dynamics of condensates formation? It is widely proved that nuclear condensates are responsive to RNA abundances and species and this pattern have not been considered in this study.

Reviewer #2

(Remarks to the Author)

The authors treated U2OS cells with Trichostatin A (TSA), known to unpack chromatin. They confirmed the effect of reduced chromatin heterogeneity and mobility reduction for GEM40 particles, synthetic condensates, Cajal bodies, and telomeres. They also found that light-induced condensate grew slower and was smaller after TSA treatment, and their phase boundary was shifted to a higher concentration. Finally, they show knocking down LaminA/C does not affect condensate phase behavior. This study asks an important question on the role of chromatin heterogeneity on nuclear condensate behavior.

However, I find many of the above findings, except changes in coarsening kinetics and phase boundary, are more of confirming results rather than revealing novel mechanisms. Other major concerns are

1) In the Abstract, the authors state, "we investigate the relationship between chromatin organization and the formation of embedded condensates in living cells, using both endogenous nuclear bodies and an engineered condensate system".

However, no formation of endogenous nuclear bodies was studied; only the mobility of Cajal bodies and telomeres were monitored after TSA treatment. Adding the effect of chromatin heterogeneity on the phase boundary, growth kinetics, and size distribution of endogenous nuclear bodies would add depth and significance to the study.

2) The effect of TSA treatment on cell cycle arresting and nuclear volume changes was not taken into consideration in the experiments.

3) A drug that increases chromatin heterogeneity would be needed to improve the robustness of the study.

4) Only one cell line is used. Major findings need to be repeated with another cell line to ensure the conclusions are not cell-line specific.

Minor concerns are:

1) Abstract: the statement "but the impact of chromatin mechanics and heterogeneity on nuclear condensates remains unclear" is misleading: several studies from this group and collaborators are on the role of chromatin on condensates. These studies are not fully explained in the Introduction in terms of what is known about the role of chromatin on nuclear condensate nucleation and growth neither, leaving the impression that this is the first work to study chromatin mechanics on condensate growth.

2) The introduction does not acknowledge that simulations have predicted the effect of local heterogeneity on phase behavior (Syles and Rosowski).

3) The paper has very detailed explanations that offer great clarity but could be modified to be more concise, particularly the introduction and the discussion.

4) Figure 2: The effect of chromatin perturbation on condensate formation and growth dynamic at other TSA treatment times (e.g. 25 hours and 13.5 hours) is not described.

5) Figure 5: Change in mode of mobility of peripheral vs interior GEM40s not convincingly linked to changes in chromatin heterogeneity. Also, throughout the paper, the mobility of interior vs peripheral particles is described but is not related to either chromatin heterogeneity or phase behavior.

6) It would be good to discuss whether endogenous chromatin heterogeneity shifts would be sufficient and timely enough to modulate biological condensates.

7) Several typos are spotted and highlighted in Word, and one comment is still in the supplemental.

Reviewer #3

(Remarks to the Author)

In the manuscript "Chromatin Heterogeneity Controls Nuclear Condensate Dynamics and Phase Behavior", the authors experimentally investigated the influences of the material properties of chromatin on the formation of condensates in the cell nucleus. With a series of experiments using both artificial and endogenous condensates, the authors demonstrated that

decreasing chromatin heterogeneity hinders the formation of condensates, both dynamically and thermodynamically. This is interpreted as an increasing stiffness of the soft regions, which suppresses phase separation. The observed behavior is consistent across several different types of condensates and ways of homogenization, suggesting the interpretation is robust.

The results are convincing and the text is written in a way that is broadly accessible to a wide audience, including us theorists. From our theoretical point of view, the manuscript provides strong evidence that condensates interact mechanically with their environment in such biological systems. Particularly interesting is the focus on the heterogeneity of the nucleus, which could affect the size and spatial distribution of condensates. I believe that the authors' timely work can inspire more experiments and also theories to model and understand biological condensates.

In summary, I am inclined to recommend its publication in Nature Communication after the following points have been addressed.

Major points:

1) Power-law functions are used in the manuscript to fit the MSD curves. However, particularly the TSA data in Fig 3b,f,j is curved quite a bit, suggesting that the MSD does not follow a power-law. Moreover, in most of the cases the time range is limited to one order of magnitude, which seems to be not sufficient for robust conclusions about the exponents and the pre-factor. For example in Fig. 3, panels (c,d) and (g,h) seem to distinguish the case of FUS Corelet from that of Cajal body by a reduction of either the exponent or the coefficient. It is more likely to me that these cases are in transitions between regimes, while the time scales of the transitions are different between cases. [Macromolecules 48, 847–862] provides an example of the scalings of diffusion of particles in polymer network. If the authors insist on a power-law fit, I would expect that the fit is shown over the entire range and not just indicated by the very short lines, which are difficult to compare to the actual data. A similar question arises around Line 260, where the exponents are used to suggest different coarsening mechanisms between control and the TSA treated cells. It seems the scalings are not robust enough to draw these conclusion. On the other hand, is it possible to track the coarsening event from the experimental images? Could different mechanisms be directly observed from the coarsening event?

2) In Fig 4a, the size of the cell nuclei seems also increase with the treatment time. Is it true or just due to visualization? If it is the case how would this contribute to phase separation?

There are also minor points that the authors might want to address:

Fig. 1g and Line 160: It would be helpful to have a line of slope 0.75 in Fig. 1g for reference.

Fig. 1h: The binning, and particularly the representation of the histogram, is very problematic. First, the bins for large Δt overlap. For small Δt , there are huge gaps, which make interpreting this histogram difficult. Consequently, I am not convinced that there is a truly bimodal distribution. Generally, it might be better to use a cumulative distribution function, which does not require binning. There might also alternative representations of the data if the claim of bimodality is indeed true.

Line 189: “[...] leading to an overall reduction in the mesh size of the chromatin network.” Do you have evidence that the mesh size changes? To me, this sounds as a hypothesis at this point and should probably be marked as such.

Line 302: Why would the size “reach a plateau” when coarsening takes place?

Line 302 and Fig. 3: It was claimed that the motion of the condensates are tracked for 1 minute, while in Fig. 3 the MSDs are only shown for 10 seconds. What is the reason for this discrepancy?

Line 325: Sentence “...we examined telomeres, which are protected ends of telomeres that assemble...” is confusing.

Line 363: The label should be (k) and (l).

Line 403 and Fig. 4: Why do you give the core-to-IDR ratios in powers of 2? Particularly in the figure, I find it difficult to get an intuitive grasp of numbers like 2^5 and 2^7 . I think powers of 10 are much more commonly used.

Fig. 4a: Is there a control study without TSA treatment showing that condensates persist? If not, one could claim that condensates disappear due to aging of the entire cell.

Fig. 4a,b: Is it possible to quantify the total amounts in the condensates and the dilute phase to check material conservation (e.g., using the data in panel b)? More generally, what is the role of production and degradation in these experiments?

Line 432: What does “both before and after blue light activation” mean? Does it mean that some of the images in panel (a) are before blue light treatment? This can be clarified.

Line 557: A particularly interesting soft matter system is the follow-up study of ref. 42: <https://doi.org/10.1038/s41563-023-01703-0>. This study shows that elastic matrices can suppress coarsening (which might explain the slower coarsening rates observed in the TSA experiments). The accompanying theory paper <https://doi.org/10.1103/PhysRevX.14.021009> suggests that heterogeneity in elastic material properties could lead to such arrest, which sounds particularly relevant in the case of chromatin.

Fig. S2(b): It seems the size correlation coefficient divides into two groups for the control. Is there a reason?

Reviewer #4

(Remarks to the Author)

Version 1:

Reviewer comments:

Reviewer #1

(Remarks to the Author)

I acknowledge that the authors performed numerous additive experiments and analyses in an attempt to address the criticisms raised.

However, there are still relevant critical points that strongly limit the biological significance, novelty, and relevance of this manuscript.

Below are some critical points:

1. The simplistic, superficial, and biologically incorrect description of chromatin as a unified polymer that transits from an undefined heterogeneous to a homogeneous state represents the rationale of this work. Indeed, the authors state from the title that "Chromatin heterogeneity controls nuclear condensates dynamics and phase behavior." Chromatin is heterogeneous as its biophysical and biochemical state is highly and dynamically regulated, both at the nano and meso-scales. In addition, the 3D genome topology demarks specific spatial 3D chromatin compartments that respond to stimuli differently: some regions can sustain a large spectrum of perturbations, while others are more sensitive and responsive. In light of this, considering that chromatin will behave homogeneously in response to a single perturbation (TSA) is incorrect, and any conclusion based on this assumption is not sustainable. In addition, the proposed methodology used to support this statement (coefficient of variation of H2A signal) is arbitrary and not supported by any biological data.
2. The strong limitation of the described approach is also highlighted by the observation by which the motion of the condensates (both the synthetic and the endogenous one) resulted in being reduced in a condition of a "homogeneous" chromatin state. These results are interpreted as the resulting spatial distribution of chromatin reduces the available space for condensation motion. However, this is mere speculation: the obtained measurements are recoding the different motions of condensates upon long-term TSA treatment, which causes many indirect effects that are not controlled nor properly considered in the interpretation of the results. Another critical point: Are these differences in mobility biologically relevant?
3. Some incongruency in the obtained results strongly limits the significance of the robustness of the performed analyses. For example, how is it possible that the Cajal bodies (whose mobility has been determined by Coilin labeling) would show a "Brownian motion" with a diffusion exponent α of 1? This measurement is inconsistent with many other MSD data referring to chromatin factors, TFs, and condensates (including the data presented in this work), considering that chromatin would constrain the motion of macromolecules within the nuclear space.
4. These aspects are further underlined by the poor statistics applied to determine the correctness of the proposed null hypothesis in each experiment. Indeed, many experiments were performed on two biological replicates. In addition, in all the experiments, the applied statistical tests consider "n" the number of objects measured/condition and not the number of biological replicates from which the values have been retrieved. By doing so, the statistical tests do not address the null hypothesis by which the measured averaged difference among the tested conditions depends on the treatment (TSA). However, the authors drew their conclusions based on the mentioned null hypothesis, which has not been tested statistically. The reported issues are well exemplified by the statistical test applied in Fig. 2e and 2f, in which the very marginal differences in the artificial biomolecular condensates diameters (Corelet system) results being statistically significant only because the authors used as "n" the number of condensates (>500) retrieved from only two biological replicates.
5. If someone would consider the correlation between transition in chromatin state (from heterogeneous to homogeneous) and condensates formation still biologically relevant (regardless of the biological inconsistencies described in point 1), then this topic will result in poor novelty, as the same group published different papers covering the same topic, including the very recent publication entitled "Chromatin compaction during confined cell migration induces and reshapes nuclear condensates".

Given the major biological issues, the inconsistency of the data, and the inappropriate statistical approaches, I do not support the publication of this manuscript in Nature Communications.

Reviewer #2

(Remarks to the Author)

The authors have addressed most of my comments except the following:

1) For the endogenous condensates: it is understandable that not all biophysical quantifications can be carried out with endogenous condensates. I suggest the authors be more specific in drawing conclusions and avoid extrapolating observations made with engineered condensates to endogenous condensates. For example, the authors state "We demonstrate that decreasing chromatin heterogeneity with epigenetic modifying drugs impairs condensate growth and mobility by shifting the binodal phase boundary." The shifting in phase boundary was not demonstrated with endogenous condensates. Also, for the nuclear speckle sizes added in Figure 2jk, unlike the engineered system, nuclear speckle formation is known to be linked to transcription. Therefore, an alternative interpretation of the size changes after TSA is because of its effect on transcription. Another way to change chromatin heterogeneity, such as that shown in Figure R2, would be needed to confirm the interpretation.

2) For the cell cycle and nuclear volume after TSA: the effect of the cell cycle on chromatin heterogeneity was addressed with new data, but the dependence of phase boundary, condensate sizes, and mobility on the cell cycle was not measured. For the volume effect, this would not be a concern for the engineered condensates where phase diagrams are mapped, but for the endogenous condensates such as newly added Figure 2jk, lowering of protein concentration due to increased nuclear volume would play a role, and this needs to be considered. Measuring eYFP-SRRM2 intensity or protein level with Western may not be sufficient as nuclear speckles, like many endogenous condensates and unlike the engineered condensates, are formed by multiple components. Again, I think perturbing the chromatin heterogeneity using a different method, such as that shown in Figure R2, would be needed to confirm that the results can be attributed to chromatin heterogeneity.

3) A drug to increase chromatin heterogeneity: the data in Figure R2 is very convincing and would address all concerns regarding the side effects of TSA outlined in points 1 and 2 above. Interestingly, the authors decide not to include the data or the method in this paper. Without such a confirmation, the side effects of TSA remain a concern for this reviewer.

Reviewer #3

(Remarks to the Author)

The revised manuscript is much improved and the authors addressed almost all comments satisfactorily. There are only a few remaining points:

1) The scalings in Fig 3 are much improved, but the lines in panels b,f,j are still not straight enough to have conclusive scaling laws in my opinion. I accept that they are sufficient for the current conclusion of the manuscript, but it might be advisable to tone down the claims about scaling laws.

2) Regarding tracking the condensates, the authors explain that there are technical difficulties for direct observation of coarsening events. However, two different mechanisms should be already distinguishable from the moving paths. Since the MSDs are calculated, the moving paths of condensates must be already obtained by tracking the positions of the condensates (e.g. in Fig. 3 a,e,j). It should be possible to check whether the condensates coarsen through Ostwald ripening, where the condensates's positions stay almost unchanged, or through Brownian motion, where the the paths of two condensates may meet and merge into one.

3) The authors might want to check the Journal information of Ref. 42, which seems to be incorrect.

4) The response to the concern of using powers of 2 in the labeling of ticks in Fig. 4 is confusing. The authors claim that "By using powers of 2, [they] can expand this range to 5 decades", which is clearly wrong, since there are still only 1.5 decades independent of the base used for the labeling. It is likely much more intuitive to label powers of 10 (even when using $10^{1.5}$ or something like that). Even better might be simple numbers or a linear axis, which should not change the presentation much, given that the logarithmic axis covers less than two decades.

Reviewer #4

(Remarks to the Author)

Version 2:

Reviewer comments:

Reviewer #2

(Remarks to the Author)

The added sorbitol data have already been published in their previous paper. Interestingly, the authors repeated the experiment here instead of referring to the published paper. Since the point was made in the previous paper, this paper has little significance in that regard. The effect on endogenous condensates is new, but the sorbitol experiment was not done on Cajal bodies or nuclear speckles.

Reviewer #3

(Remarks to the Author)

The authors addressed my specific comments satisfactorily. Concerning the comments of the other referees, I agree with them and think that the authors might over-interpret their results. As far as I can see, the authors essentially showed a correlation between chromatin heterogeneity and the behavior of condensates, but the title and abstract imply causation. It is unclear from the provided data whether heterogeneity or any other aspect that might be changed by the treatment (e.g., overall stiffness or chemical properties) is responsible for the observed behavior. However, the observed correlation is definitely interesting and points toward a cross-talk between condensates and their environment. I thus believe this work should be published if causation is no longer implied or supporting evidence for it is added.

Reviewer #4

(Remarks to the Author)

REVIEWER COMMENTS

We sincerely thank all the reviewers for their valuable feedback and constructive suggestions, which have been instrumental in enhancing the quality of our manuscript. In response to the reviewer's comments, we have conducted a significant number of new experiments and analysis, and included additional data in both the main manuscript and the supplementary document. These efforts have allowed us to address the reviewer's concerns comprehensively and have greatly improved the robustness and clarity of our findings. We believe that these enhancements have strengthened the overall impact of our study, and we are grateful for the opportunity to refine our work with the reviewer's guidance.

Reviewer #1 (Remarks to the Author):

In this work, Xia and collaborators investigated the contribution of chromatin heterogeneity to the dynamic formation of chromatin condensates and phase separation shift. Specifically, they considered the potential relevance of the local mechanical properties of chromatin as a modulator for condensates dynamics through phase separation. They further extend their results proposing that the modulation of chromatin heterogeneity may represent a regulatory mechanism to control phase transition, thus impacting on condensates formation. Although the forwarded biological question is of high relevance as it proposed a central role for the local mechanical forces in regulating nuclear condensates, the experimental design and the provided data are not adequate to properly address this question. The results obtained do not support the raised conclusions, and the authors need to perform a series of controls to strengthen the robustness of the results and the significance of their data. In addition, the work contained multiple approaches and experimental settings which are poorly controlled, which reduced the interest and the relevance of the findings.

In sum, although the proposed role of the local chromatin mechanics in tuning phase separation of nuclear condensates is of high relevance, further experiments are required to improve the soundness of the raised conclusions. Indeed, there are important technical limitations and a lack of appropriate controls that reduce the impact and the robustness of this study that need to be addressed before considering the manuscript ready for publication.

Major criticisms:

1. The authors used a poorly controlled drug-based approaches to impact the chromatin states, namely the usage of the HDAC inhibitor TSA. It is highly documented how this perturbation can affect many aspects of chromatin functions and cellular responses, yet the author did not perform any control to determine the cellular, transcriptional or chromatin response, apart detecting the intensity of the miRFP670-H2B and retrieving an indirect measurement of chromatin heterogeneity. They used a single cell line, in which they transiently overexpressed all the biosensor used in the manuscript, never showing the level of heterogeneity nor the expression level. For example, what is the level of the overexpressed H2B, with respect to the endogenous protein? Is it miRFP670-H2B organized into nucleosomes incorporated into the chromatin. What are the consequences of its over-expression? Having stable clones expressing a quasi-physiological level of the reported is mandatory to reach any conclusion of this work. To better control the system, a knock-in of the miRFP670- into one of the endogenous loci of H2B will ensure the correct measurement of chromatin distribution, in response to any perturbation. Without these controls in at least two different cell lines no reliable conclusion can be raised by this study.

We appreciate the reviewer's suggestions. We agree that additional control experiments will significantly enhance the robustness of our findings. Therefore, we have conducted new control experiments and added the corresponding results to the manuscript and supplementary material. Additionally, we concur that repeating the main findings in another cell line will further strengthen our results. Consequently, we have repeated the key experiments in HeLa cells and included these results in the manuscript.

To generate the miRFP670-H2B U2OS stable cell line, we used lentiviral transduction instead of transient transfection. To quantify the overexpression levels of miRFP670-H2B and endogenous H2B in U2OS cells, we performed western blot on both wild-type and miRFP670-H2B U2OS cells. The results show that transduced cells exhibit two bands when stained with an anti-H2B antibody: the lower molecular weight band (~14 kDa) corresponds to endogenous H2B, and the higher molecular weight band (~55 kDa) corresponds to overexpressed miRFP670-H2B. In contrast, wild-type U2OS cells display only the lower molecular weight band (~14 kDa), corresponding to H2B, as shown in **Supplementary Fig. S1(a)**. To confirm that the higher molecular weight band corresponds to miRFP670-H2B, we conducted a Western blot on both cell lines using an anti-miRFP antibody. The results show that only the miRFP670-H2B cells exhibit the higher molecular weight band (~55 kDa), while wild-type cells do not, confirming that the higher molecular weight band corresponds to miRFP670-H2B, as shown in **Supplementary Fig. S1(b)**. Additionally, we quantified the expression level of H2B by measuring the integrated intensity of each band in **Fig. S1(a)**. We show that miRFP-H2B transduced cells only have approximately 5% higher total H2B expression compared to wild-type cells, as shown in **Supplementary Fig. S1(c)**.

To investigate the effects of this small amount of H2B overexpression and determine whether it is toxic to the cells or adversely affects cellular response, we conducted new experiments comparing several aspects of cell behavior between wild-type and transduced cells. These experiments assessed cell viability, cell proliferation, and cell cycle distribution. We performed a live/dead assay to measure the viability of wild-type and transduced U2OS cells, as shown in **Supplementary Fig. S2(a)**. We found no significant difference in viability between the wild-type and transduced U2OS cells, as shown in **Supplementary Fig. S2(b)**. We also quantified the growth curve of wild-type and transduced U2OS cells using a commercially available colorimetric cell proliferation assay. During the 3-day culture period, there was no observable difference in the growth curves of the two cell types, as shown in **Supplementary Fig. S3**. Additionally, we transduced both cell types with the Fluorescent Ubiquitination-based Cell Cycle Indicator (FUCCI) construct to monitor the cell cycle over 36 hours. Cells express different fluorescent proteins at various stages of the cell cycle: in the S/G2/M phase, cells express a monomeric Azami Green reporter (green); in the G1 phase, cells express a monomeric Kusabira Orange 2 reporter (red); and during the transition from G1 to S phase, cells express both GFP and RFP reporters, as shown in **Supplementary Fig. S4(a)**. We quantified the cell cycle every 12 hours over the 36-hour period and did not observe any differences between wild-type and miRFP670-H2B-transduced cells, as shown in **Supplementary Fig. S4(b)**.

To verify that miRFP670-H2B is incorporated into the chromatin in U2OS cells, we performed fluorescence recovery after photobleaching (FRAP) on two components of chromatin: miRFP670-H2B and DNA in live cells, as shown in **Supplementary Fig. S5(a)**. DNA was stained with Hoechst. Our results show that the immobile fraction of miRFP670-H2B is 3%, which is even lower than the immobile fraction of the DNA compartment, which is 15%, as shown in **Supplementary Fig. S5(b)**. This suggests that 97% of miRFP670-H2B is incorporated into the chromatin of U2OS cells, rendering it immobile. Additionally, several studies have used imaging-based methods to quantify chromatin density and its spatial variations¹⁻⁶. Thus, we believe that this imaging-based approach is feasible for quantifying the spatial distribution and heterogeneity of the chromatin network.

Our results show that transducing the miRFP670-H2B construct into cells induces only a small amount of overexpression, as demonstrated by our Western blot results. We also confirmed that majority (>95%) of miRFP670-H2B is incorporated into the chromatin. Furthermore, the introduction of this construct does not cause any observable differences in cell behavior, as supported by our new experiments quantifying various cell behaviors. Taken together, we believe that our system utilizing miRFP-H2B expression is reflecting quasi-physiological behavior of H2B incorporated into chromatin.

We also agree that repeating the main experiment in a different cell line would significantly strengthen the robustness of our findings. Therefore, we conducted a series of new experiments using HeLa cell lines with appropriate controls and confirmed similar results.

We introduced a miRFP670-tagged (miRFP-) H2B construct into HeLa cells and verified nearly complete incorporation of miRFP-H2B using fluorescence recovery after photobleaching (FRAP) (**Supplementary Fig. S5(c) and (d)**), with minimal impact on total H2B expression levels as determined by Western blot quantifications

(Supplementary Fig. S1(d), (e), and (f)) and low toxicity following lentiviral transduction (Supplementary Fig. S2(c) and (d)). Consistently, we observed a similar decrease in chromatin heterogeneity when HeLa cells were treated with TSA (Supplementary Fig. S8(a) and (b)). To investigate phase behavior, we introduced Corelet-engineered condensates into HeLa cells and used blue light activation to trigger phase separation. Upon TSA treatment, we observed a reduction in condensate size and a shift in their size distribution (Supplementary Fig. S11(a), (b), (c), and (d)). Additionally, mapping the phase diagram of these condensates revealed a shift in the phase boundary towards higher core concentrations, indicating suppressed phase separation (Supplementary Fig. S16(a), (b), (c), (d), and (e)). These findings suggest that TSA-induced homogenization of the chromatin network inhibits phase separation and reduces condensate size, regardless of the specific cell line used.

2. The usage of TSA or DZNep inhibitors per se does not imply an “homogenization” of chromatin as stated by the authors. These compounds affect different kinds of chromatin compartments, with TSA causing a general uncontrolled increase of histone acetylation, effecting both eu- and constitutive (H3K9me2/3) heterochromatin. DZNep instead is a more specific inhibitor of EZH2, which deposit H3K27me1, me2 and me3, the latter being involved in Polycomb functioning in maintaining a repressive chromatin environment. To simplify the consequences of the usage of these compounds to a poorly defined and characterized “homogenization” of chromatin is at the best superficial and inadequate. The authors need to characterize the alterations of the chromatin states by imaging and sequencing, after having ensured that the toxicity of the treatment did not alter the cellular responses (vitality, cell cycle progression, cell adhesion ect. They further should clarify the reasons why they chose these drugs and the rationale behind comparing such a diverse chromatin pattern in the investigated system.

We appreciate the reviewer's questions. We acknowledge that these drugs act at the molecular level and have distinct mechanisms of action. However, the heterogeneity quantified in this paper is at the mesoscale, and we apologize for the confusion and not clearly defining “heterogeneity” in our study. We have added clarifications in the manuscript to address this issue.

In our study, “heterogeneity” refers specifically to the variation in spatial density of the chromatin network at the mesoscale, as opposed to atomic or single-molecule levels of chromatin organization. It is important to recognize that the chromatin network is a hierarchical structure spanning several orders of magnitude in length, from nanometer scales, such as individual nucleosomes, to mesoscale features like Topologically Associating Domains (TADs), and up to macroscale structures like the entire chromatin network. Similarly, the mechanical properties of the nucleus also depend on the length scale being measured^{7–10}. Mesoscale heterogeneity in chromatin organization arises from the differential compaction states of chromatin, which are related to differential epigenetic modifications. Since most nuclear condensates exist at the mesoscale, typically hundreds of nanometers, we characterize chromatin heterogeneity at a similar length scale to obtain the most relevant measurements. This mesoscale also falls within the optical resolution of a confocal microscope and can thus be directly observed and quantified.

The rationale for choosing drugs to decondense the chromatin network involves either increasing histone acetylation (using TSA) or decreasing histone methylation (using DZNep) in order to broadly affect the epigenetic balance and lead to differential chromatin organization at the mesoscale. At the molecular level, these actions reduce the positive charge on histones, decreasing their affinity for negatively charged DNA and resulting in a more open chromatin structure at both the nanoscale and mesoscale. TSA inhibits histone deacetylases (HDACs), increasing histone acetylation in both euchromatin and heterochromatin regions¹¹. DZNep inhibits histone methylation, particularly at sites like H3K27me1, me2, me3, and H4K20me3, leading to chromatin decompaction¹².

At the microscale level of a few nanometers, studies have shown that both drugs increase the accessibility of molecular machinery to chromatin^{13,14}. Consistently, at the mesoscale level of hundreds of nanometers, several studies have demonstrated that TSA leads to a more homogeneous spatial distribution of the chromatin network.^{11,15,16} Specifically, we have referenced a study that illustrates changes in chromatin microstructure

before and after TSA treatment using super high-resolution Transmission Electron Microscopy (TEM), as shown in **Fig. R1**¹⁶. These studies consistently show that TSA results in a more homogeneous chromatin network structure at the mesoscale, which aligns well with our observations (**Fig. 1c**). Additionally, we observe consistent alteration of phase diagrams with these two different epigenetic modifiers (TSA and DZNep), which bolsters our confidence that the phase diagrams are altered because of mesoscale chromatin mechanical changes rather than, for example, acetylation to the FUS protein itself (**Fig. 4**).

We also investigated the effects of these drugs on cell behavior. Compared to untreated cells, DZNep caused a slight inhibition of cell growth, while TSA resulted in 12% cell death after 32 hours (**Supplementary Fig. S19**). However, unhealthy cells were excluded from our analysis by examining cell morphology for signs of shrinkage, bleb formation, or irregular nuclear morphology. For cell cycle analysis, we used FUCCI-transduced cells and observed a 13% increase in the G₁ phase population and a 13% decrease in the G₂/S phase population after 36 hours of TSA treatment. DZNep caused a 13% increase in the G₁ phase population with no change in the G₂/S phase population (**Supplementary Fig. S4(c)**). Cells undergoing mitosis are rarely observed, likely because mitosis is much shorter than interphase. We investigated chromatin heterogeneity across different stages of interphase but found no statistically significant differences, suggesting that mesoscale chromatin heterogeneity is unlikely affected by cell cycle stage variations in interphase (**Supplementary Fig. S20**). Therefore, we do not expect these minor cell cycle shifts to significantly impact the phase behavior observed in our study. Additionally, we excluded cells in mitosis from our analysis since their chromatin morphology is entirely different and highly condensed. These measures ensure that our results are minimally affected by small populations of unhealthy cells and slight cell cycle shifts.

Figure R1. Transmission electron microscopy of chromatin network before and after TSA treatment

Transmission electron microscopy images collected by ESI. Phosphorus elemental maps of 50-nm ultrathin sections show a nucleus of a C3H/10T1/2 cell (left) and a nucleus of a cell treated with 100 nM TSA for 24 h (right). The circles on the bottom right of the -TSA image are 30 nm and 100 nm in diameter. This figure is from Strickfaden, H., Tolsma, T.O., Sharma, A., Underhill, D.A., Hansen, J.C. and Hendzel, M.J., 2020. Condensed chromatin behaves like a solid on the mesoscale in vitro and in living cells. *Cell*, 183(7), pp.1772-1784.

3. The authors interpreted the obtained results, proposing that the uncontrolled perturbation of the chromatin states results in a relevant change of the local mechanical forces that. Impinges on the phase separation and the assembly of condensates. Unfortunately, they did not provide any direct or indirect measurements of chromatin mechanics in any of the artificial biological contexts.

We apologize for not explaining this more clearly in the initial submission of our manuscript. We have added more explanations to avoid potential confusion for readers.

We measured the mobility of GEM40 nuclear probes as a direct measure of the mesh size, a feasible approach as demonstrated by the Holt lab and others¹⁷⁻¹⁹. Mesh size was then used to infer local mechanics, given the intrinsic link between mesh size and stiffness in polymer networks, where smaller mesh sizes typically correspond to stiffer networks^{20,21}. Using GEM40 probes, we quantified their mobility to assess changes in chromatin mesh size following TSA treatment (**Fig. 1(d)-(i)**). Our results show that TSA treatment significantly reduces GEM40 mobility (**Fig. 1(i)-(l)**), indicating a decrease in chromatin mesh size and suggesting an increase in local chromatin stiffness.

4. All the biophysical measurements appear to lack the appropriate controls, ensuring the robustness of the raised conclusions. For example, although the high variability associated with transient expression of the used biosensor, the authors did not provide biological replicates. Moreover, they never considered the possible perturbation of the cell cycle state, nor considered that cells in different stages of the cell cycle are characterized by important difference in terms of 3D chromatin organization, histone modifications, and chromatin compaction. How do the authors interpret their data considering that on average a small fraction of the nuclear volume is occupied by the nucleosome in interphase?

We again appreciate the thoughtful feedback. We agree that some of the results initially lacked information on biological replicates. We have now included this information in the figures. All of the experiments have been repeated at least twice with similar results; some are repeated more than three times.

We agree that the cell cycle can lead to varying levels of chromatin compaction. The most pronounced change occurs during the mitotic phase, where chromatin forms super-compacted chromosomes. We excluded these cells from our data analysis due to their distinct chromatin organization and focused only on cells in interphase. In fact, cells undergoing mitosis are rarely observed, likely because mitosis is much shorter than interphase. For interphase cells, we observed a small shift in the cell cycle distribution. Specifically, we observed a 13% increase in the G₁ phase population and a 13% decrease in the G₂/S phase population after 36 hours of TSA treatment. DZNep caused a 13% increase in the G₁ phase population with no change in the G₂/S phase population (**Supplementary Fig. S4(c)**). We investigated chromatin heterogeneity across different stages of interphase but found no statistically significant differences, suggesting that mesoscale chromatin heterogeneity is unlikely affected by cell cycle stage variations in interphase (**Supplementary Fig. S2o**). Therefore, we do not expect these minor cell cycle shifts to significantly impact the phase behavior observed in our study.

Chromatin and lamin have been recognized as two key mechanical elements of the nucleus^{9,10,22}. Chromatin is distributed throughout the nucleus, while lamin is located beneath the nuclear membrane. Despite occupying a relatively small volume percentage of the nucleus, chromatin plays a dominant role in determining its internal mechanical properties⁴⁴. Unlike protein solutions, which occupy a large volume percentage of the nucleus but are liquid and cannot provide mechanical support, chromatin forms a 3D network that acts as a stiff polymeric gel²³. This viscoelastic network provides partially solid-like properties that offer mechanical support within the nucleus. Therefore, the structure of the chromatin network is crucial for determining its mechanical properties.

5. All the imaging measurements have been performed in 2D, without taking into consideration the spatial (3D)

genome organization, especially of heterochromatin, which is characterized by long-range interactions clustering together non-linear portion of the genome. The Authors should perform the imaging-based measurements in 3D.

We agree that quantifying chromatin distribution in 3D can provide a more comprehensive representation. Therefore, we conducted additional experiments to image the chromatin structure in a z-stack (3D) before and after TSA treatment. We found that the results remained consistent, with TSA decreasing chromatin heterogeneity in 3D (**Supplementary Figure S7**).

6. The proposed models are based on an experimental design that investigate the behavior of heterogenous, exogenous proteins. Although the designed tools permit to overcome some intrinsic limitation in measuring the biophysical properties of condensates within the cellular context, the authors did not provide and substantial data demonstrating that the measured behavior represent a general trend of the endogenous counterpart. The inclusion of a reporter of Cajal body or for the telomere based of the over-expression of a reporter system does not entail the authors to extend their conclusion to the endogenous condensates.

We thank the reviewer for the suggestion. We agree that adding an endogenously tagged cell line will be beneficial to extend the broader implications of our findings. We performed new experiments using a HEK cell line with CRISPR-Cas9-based tagging of eYFP at the SRRM2 gene locus³¹, which can be used to visualize endogenous nuclear speckles. We imaged the endogenous tagged nuclear speckles before and after chromatin homogenization with TSA (**Fig. 2(j)**). We demonstrate that the size of the nuclear speckles significantly decreases after the chromatin network is homogenized by TSA, while the control group with no drug shows no change in size (**Fig. 2(k)**). This result is in good agreement with our other findings using the Corelet condensates, which suggests the finding is universal for both model Corelet condensates and endogenous condensates.

7. The perturbation of the chromatin states with these compounds will certainly also impact the gene expression state and transcripts abundances. What is the impact of altering the expression rates on the dynamics of condensates formation? It is widely proved that nuclear condensates are responsive to RNA abundances and species and this pattern have not been considered in this study.

We agree that the interaction of RNA with condensates is an interesting and certainly important question. However, in this study, to simplify the interpretation of experiment results and demonstrate the proof of concept, we use model condensates with simplified version of IDR with RNA recognition motif (RRM) truncated³³⁻³⁵, for example, FUS IDR(1-214) and HNRNPA1 IDR(186-320). Due to the lack of RRM domain, these model condensates are less likely to interact with the RNA present in the cell. And the behaviors of the model condensate are able to recapitulate those of nuclear condensates, as demonstrated in this manuscript (dynamics of Cajal body, size distribution of nuclear speckles). Therefore, we reason that the chromatin heterogeneity is the major contributor to change the phase behavior and dynamics of condensates in the nucleus.

We have further clarified this in the manuscript; it now reads: "Importantly, we note that the IDR segments we used (FUS, HNRNPA1) are truncated versions of the full-length proteins, lacking the RNA recognition motif (RRM) domain and thus are less likely to be impacted by RNA concentrations in the cell."

Reviewer #2 (Remarks to the Author):

The authors treated U2OS cells with Trichostatin A (TSA), known to unpack chromatin. They confirmed the effect of reduced chromatin heterogeneity and mobility reduction for GEM40 particles, synthetic condensates, Cajal bodies, and telomeres. They also found that light-induced condensate grew slower and was smaller after TSA treatment, and their phase boundary was shifted to a higher concentration. Finally, they show knocking down LaminA/C does not affect condensate phase behavior. This study asks an important question on the role of chromatin heterogeneity on nuclear condensate behavior. However, I find many of the above findings, except changes in coarsening kinetics and phase boundary, are more of confirming results rather than revealing novel mechanisms.

We thank the reviewer for their thoughtful feedback and suggestions to improve our study. In response, we have conducted a large number of additional supporting experiments and included the results in the revised version of the manuscript.

We appreciate the reviewer's recognition of the importance of this topic. We would also like to clarify that while size distribution has been explored in the unperturbed nucleus, the impact of chromatin heterogeneity has not been previously examined. In our study, we focused on modulating the heterogeneity of the chromatin network, and demonstrated that chromatin heterogeneity can influence the size distribution, growth dynamics of nuclear condensates, as well as affect phase equilibrium and shift the binodal phase boundary.

Other major concerns are

1) In the Abstract, the authors state, "we investigate the relationship between chromatin organization and the formation of embedded condensates in living cells, using both endogenous nuclear bodies and an engineered condensate system". However, no formation of endogenous nuclear bodies was studied; only the mobility of Cajal bodies and telomeres were monitored after TSA treatment. Adding the effect of chromatin heterogeneity on the phase boundary, growth kinetics, and size distribution of endogenous nuclear bodies would add depth and significance to the study.

In our study, we have focused on utilizing a model engineered condensate system (Corelets), with less focus on endogenous condensates, due to the challenge of performing similarly careful biophysical measurements on them, for example controlling biomolecular concentrations and externally triggering phase separation behavior.

However, we recognize the importance of bringing the behavior of endogenous condensates more centrally into our study to examine the generality of our findings. We therefore conducted new experiments to investigate the size of endogenously tagged nuclear speckles (eYFP-SRRM2) in the HEK cell line generated using CRISPR-Cas9. We compared the size of these nuclear speckles before and after chromatin homogenization with TSA (**Fig. 2(j)**). Our results demonstrate that the size of nuclear speckles significantly decreases after the chromatin network is homogenized by TSA, whereas the control group without drug treatment shows no change in size (**Fig. 2(k)**). This result aligns well with our findings using the model Corelet condensates, suggesting that this phenomenon is universal for both engineered and endogenous condensates.

2) The effect of TSA treatment on cell cycle arresting and nuclear volume changes was not taken into consideration in the experiments.

We acknowledge that the cell cycle can result in different levels of chromatin compaction. The most significant change occurs during the mitotic phase, where chromatin forms highly compacted chromosomes. We excluded these cells from our data analysis due to their distinct chromatin organization and focused solely on interphase

cells. In fact, cells undergoing mitosis are rarely observed, likely because mitosis is much shorter than interphase. In interphase cells, we noted a slight shift in the cell cycle distribution. Specifically, there was a 13% increase in the G₁ phase population and a 13% decrease in the G₂/S phase population after 36 hours of TSA treatment. DZNep treatment led to a 13% increase in the G₁ phase population with no change in the G₂/S phase population (**Supplementary Fig. S4(c)**). We investigated chromatin heterogeneity across different stages of interphase but found no statistically significant differences, suggesting that mesoscale chromatin heterogeneity is unlikely affected by cell cycle stage variations in interphase (**Supplementary Fig. S2o**). Therefore, we do not expect these minor cell cycle shifts to significantly impact the phase behavior observed in our study.

We performed a new experiment to measure the nuclear volume of cells and found an approximate 20% increase after 38.5 hours of TSA treatment (**Supplementary Fig. S6**). This increase occurs because chromatin decompaction allows the chromatin fibers to occupy a larger volume within the nucleus. As dense chromatin regions decompact, they expand into chromatin-sparse regions and lead to an overall increase in nuclear volume. We attribute this nuclear volume increase to chromatin decompaction. Similarly, changes in phase behavior result from the homogenization induced by chromatin decompaction. Therefore, we do not observe a direct cause-and-effect relationship between nuclear volume and phase behavior or condensate dynamics; rather, both are outcomes of chromatin decompaction and homogenization. Additionally, we directly measure the real-time concentration of proteins using quantitative fluorescence microscopy, which is unaffected by changes in nuclear volume and thus accurately maps the phase diagram.

3) A drug that increases chromatin heterogeneity would be needed to improve the robustness of the study.

We thank the reviewer for the suggestion. We attempted to use methylstat, a histone demethylase inhibitor, but did not observe significant changes in chromatin heterogeneity through imaging. We reason that while methylstat increases specific heterochromatin markers like H₃K₉me_{2/3} and H₃K₂₇me₃, the overall chromatin distribution may not change visibly.

However, we note that we have observed consistent results using another approach to vary chromatin heterogeneity, which is described in still unpublished manuscript recently submitted to Nat. Comms. from our lab (entitled 'Chromatin Compaction During Confined Cell Migration Induces and Reshapes Nuclear Condensates', in revision). In this work, we applied hyperosmotic treatment to enhance chromatin network heterogeneity (**Fig. R2(a) and (b)**). We measured the phase diagram using the Corelet system with FUS IDR in living cells and found that the binodal phase boundary shifted upward, indicating promoted phase separation (**Fig. R2(c), (d), and (e)**). This suggests a positive correlation between chromatin network heterogeneity and phase separation promotion, aligning with our current manuscript findings.

Figure R2. Chromatin heterogeneity tunes nuclear condensates phase separation in MDA-MB231 cells

(a). Representative images of MDA-MB231 cells expressing miRFP670-H2B under sorbitol treatment. (b). Scatterplots show the difference in chromatin heterogeneity before and after sorbitol treatment. *** $p < 0.001$ according to one-way ANOVA test (two-tailed) from 4 independent technical replicates for each condition. (c). Phase boundary measured in cells expressing Corelet condensates before sorbitol treatment. (d). Phase boundary measured in cells expressing Corelet condensates after sorbitol treatment. (e). Comparison of phase boundary before and after sorbitol treatment. $n > 100$ cells for each experiment.

4) Only one cell line is used. Major findings need to be repeated with another cell line to ensure the conclusions are not cell-line specific.

We thank the reviewer for their suggestions. We agree that repeating the major experiment in another cell line would greatly enhance the robustness of our findings. To this end, we conducted a series of new experiments with HeLa cell lines with appropriate controls and confirmed that similar results were observed.

We introduced a miRFP670-tagged (miRFP-) H2B construct into HeLa cells and confirmed the nearly complete incorporation of miRFP-H2B using fluorescence recovery after photobleaching (FRAP) (Supplementary Fig. S5(c) and (d)), with minimal disturbance to total H2B expression levels by Western blot quantifications (Supplementary Fig. S1(d), (e) and (f)) and low toxicity to the cells following lentiviral transduction (Supplementary Fig. S2(c) and (d)). Consistently, we observed a similar decrease in chromatin heterogeneity when HeLa cells were treated with TSA (Supplementary Fig. S8(a) and (b)). To investigate phase behavior, we

introduced Corelet-engineered condensates into HeLa cells and used blue light activation to trigger phase separation. Upon TSA treatment, we observed a reduction in condensate size and a shift in their size distribution (**Supplementary Fig. S11(a), (b), (c) and (d)**). Additionally, mapping the phase diagram of these condensates showed a shift in the phase boundary towards higher core concentrations, indicating suppressed phase separation (**Supplementary Fig. S16(a), (b), (c), (d), and (e)**). These findings suggest that TSA-induced homogenization of the chromatin network inhibits phase separation and reduces condensate size, independent of the specific cell line used.

Minor concerns are:

1) Abstract: the statement “but the impact of chromatin mechanics and heterogeneity on nuclear condensates remains unclear” is misleading: several studies from this group and collaborators are on the role of chromatin on condensates. These studies are not fully explained in the Introduction in terms of what is known about the role of chromatin on nuclear condensate nucleation and growth neither, leaving the impression that this is the first work to study chromatin mechanics on condensate growth.

We again appreciate these comments. We have changed the description in the abstract, it now reads: “From studies in non-living materials, the driving forces for phase separation are expected to be sensitive to the local mechanical environment, which often exhibits significant spatial heterogeneity. However, the impact of chromatin heterogeneity on the phase equilibrium and dynamics of nuclear condensates remains unclear.”

In addition, we have added further explanation in the introduction to clarify this point.

2) The introduction does not acknowledge that simulations have predicted the effect of local heterogeneity on phase behavior (Syles and Rosowski).

We greatly appreciate and acknowledge the work from the Dufresne lab and others in this field, which we have cited extensively in the manuscript including the one suggested by the reviewer. However, we recognize that our initial phrasing may not have been clear enough. We have revised the sentence in the introduction to better reflect this suggestion. It now reads: “A number of recent studies using *in vitro* polymer systems and simulations suggest that the mechanical properties of the condensate environment could influence the phase equilibrium of condensates^{44–52}. In particular, mechanical heterogeneity of the substrate has been shown to affect phase separation, both *in vitro*^{44,45,53} and theoretically⁵².”

3) The paper has very detailed explanations that offer great clarity but could be modified to be more concise, particularly the introduction and the discussion.

We again appreciate the feedback. We have revised the introduction and conclusion to make them more concise.

4) Figure 2: The effect of chromatin perturbation on condensate formation and growth dynamic at other TSA treatment times (e.g. 25 hours and 13.5 hours) is not described.

We have now included data for the condensate formation at other TSA treatments (16h and 24h) for both 15s activation and 3min activation (**Supplementary Fig. S10(a) and (b)**). Our results show that condensate size consistently decreases with longer TSA treatment times (**Supplementary Fig. S10(c) and (d)**).

5) Figure 5: Change in mode of mobility of peripheral vs interior GEM40s not convincingly linked to changes in

chromatin heterogeneity. Also, throughout the paper, the mobility of interior vs peripheral particles is described but is not related to either chromatin heterogeneity or phase behavior.

Part of the spatial heterogeneity is the difference of the chromatin density in the nuclear periphery and interior. Chromatin at the nuclear periphery tends to be denser and is often associated with more compact heterochromatin, and the interior typically has lower chromatin density compared to the periphery and is mostly associated with more opened euchromatin⁴⁵. In addition to quantifying chromatin heterogeneity using microscopy, we use GEM 40 particle tracking to measure the microstructure of the chromatin network and its mechanics. We show that upon TSA treatment, the dynamics of GEM40 in the nuclear interior has been suppressed, indicating a decrease of nuclear interior mesh size, where dynamics of GEM40 in the nuclear periphery has increased, indicating an increase of interior mesh size, both the mesh size of the interior and periphery will meet in the middle, as suggested by GEM40 dynamics in **Fig. 1 (j), (k), and (i)**. By contrast, upon Lamin knockdown, the mesh size of the nuclear interior remains unchanged, as indicated by little change in the GEM40 dynamics, as shown in **Fig. 5 (e) and (f)**. This suggests minimal change in heterogeneity and is consistent with the observed no change in phase behavior. We have added further explanation in the manuscript to clarify this point.

6) It would be good to discuss whether endogenous chromatin heterogeneity shifts would be sufficient and timely enough to modulate biological condensates.

We appreciate this suggestion and have added a discussion to the manuscript. Changes in chromatin organization have been observed during stem cell differentiation, which occurs over a timescale of days. As stem cells differentiate, chromatin transitions from a more open state to a more condensed state, associated with the silencing of pluripotency genes and the activation of lineage-specific genes^{3,36,37}. Additionally, chromatin organization is known to respond to mechanical cues from the environment; stiff matrices have been shown to promote chromatin decompaction and increased accessibility, with these changes observed over a timescale of two days^{38,39}. In our assay, changes in the chromatin network occur over a timescale of 24-30 hours, which is similar to the chromatin organization changes observed during stem cell differentiation and mechanosensing of matrix stiffness.

We have added these discussions to the manuscript.

7) Several typos are spotted and highlighted in Word, and one comment is still in the supplemental.

Thank you for pointing this out. We have corrected the typos and removed the comments.

Reviewer #3 (Remarks to the Author):

In the manuscript "Chromatin Heterogeneity Controls Nuclear Condensate Dynamics and Phase Behavior", the authors experimentally investigated the influences of the material properties of chromatin on the formation of condensates in the cell nucleus. With a series of experiments using both artificial and endogenous condensates, the authors demonstrated that decreasing chromatin heterogeneity hinders the formation of condensates, both dynamically and thermodynamically. This is interpreted as an increasing stiffness of the soft regions, which

suppresses phase separation. The observed behavior is consistent across several different types of condensates and ways of homogenization, suggesting the interpretation is robust.

The results are convincing and the text is written in a way that is broadly accessible to a wide audience, including us theorists. From our theoretical point of view, the manuscript provides strong evidence that condensates interact mechanically with their environment in such biological systems. Particularly interesting is the focus on the heterogeneity of the nucleus, which could affect the size and spatial distribution of condensates. I believe that the authors' timely work can inspire more experiments and also theories to model and understand biological condensates.

In summary, I am inclined to recommend its publication in Nature Communication after the following points have been addressed.

We thank the reviewer for the thorough and encouraging feedback on our manuscript. We greatly appreciate your recognition of our experimental findings and particularly the broader accessibility of the text.

Major points:

1) Power-law functions are used in the manuscript to fit the MSD curves. However, particularly the TSA data in Fig 3b,f,j is curved quite a bit, suggesting that the MSD does not follow a power-law. Moreover, in most of the cases the time range is limited to one order of magnitude, which seems to be not sufficient for robust conclusions about the exponents and the pre-factor. For example in Fig. 3, panels (c,d) and (g,h) seem to distinguish the case of FUS Corelet from that of Cajal body by a reduction of either the exponent or the coefficient. It is more likely to me that these cases are in transitions between regimes, while the time scales of the transitions are different between cases. [Macromolecules 48, 847–862] provides an example of the scalings of diffusion of particles in polymer network. If the authors insist on a power-law fit, I would expect that the fit is shown over the entire range and not just indicated by the very short lines, which are difficult to compare to the actual data. A similar question arises around Line 260, where the exponents are used to suggest different coarsening mechanisms between control and the TSA treated cells. It seems the scalings are not robust enough to draw these conclusion. On the other hand, is it possible to track the coarsening event from the experimental images? Could different mechanisms be directly observed from the coarsening event?

We appreciate this suggestion. We agree with the caveats around interpreting power law-like behavior over finite data ranges, and that probing the full range of time scales for fitting would make the analysis more convincing. To address this, we reanalyzed the MSD for nearly two decades of lag time for Fig. 3(b), (f) and (j), which is the maximum duration we can probe without causing significant photobleaching, and updated the figures and descriptions in the manuscript.

The condensates we measured, including Corelet and Cajal bodies, typically range from a few hundred nanometers to one micron in size, which is an order of magnitude larger than the chromatin network's mesh size of approximately 40 nm. Therefore, it is unlikely that the condensates hop between the chromatin mesh; rather, they are trapped and influenced by the local mechanical properties of their surrounding environment³². For both Corelet condensates and Cajal bodies, the MSD follows a nearly straight line across two decades of time lag (Fig. 3(b) and (f)), indicating that power-law fitting is likely adequate to capture their dynamics. For Corelet condensates, there are statistically significant decreases in both the diffusion coefficient and exponent after cells are treated with TSA (Fig. 3(c) and (d)). Similarly, for Cajal bodies, there is a decrease in the diffusion coefficient, while the diffusion exponent remains unchanged (Fig. 3(g) and (h)). We conclude that TSA-induced homogenization of the chromatin network decreases the dynamics of nuclear condensates. It is also noteworthy that the diffusion exponent of Cajal bodies is larger than that of Corelet condensates. This difference is likely because Cajal bodies are endogenous condensates that actively interact with cellular machinery and are subject to active forces in the nucleus, leading to faster dynamics compared to the 'inert' engineered Corelet condensates. Similarly, we reanalyzed the mean squared displacement (MSD) for TRF2 over nearly two decades of lag time and

found that the MSD follows an almost straight line throughout this period (**Fig. 3(j)**). The diffusion exponent of TRF2 in control cells is higher than in TSA-treated cells. Additionally, the diffusion coefficient of TRF2 is greater in control cells compared to TSA-treated cells (**Fig. 3(k) and (l)**).

We appreciate the reviewer's feedback regarding the coarsening dynamics. While we agree that tracking individual condensates over the entire time course would be ideal, it is technically challenging due to their tendency to move out of focus and the difficulty in tracking their motion in 3D. Nonetheless, a recent preprint published last month is consistent with our observation of dampened droplet growth (Banerjee et.al. 2024). Therefore, we believe that the homogenization caused by TSA is likely to decrease the growth rate of condensates.

2) In Fig 4a, the size of the cell nuclei seems also increase with the treatment time. Is it true or just due to visualization? If it is the case how would this contribute to phase separation?

We conducted a new experiment to measure the nuclear volume of cells and observed an approximate 20% increase after 38.5 hours of TSA treatment (**Supplementary Fig. S6**). This increase is due to chromatin decompaction, which allows chromatin fibers to occupy more space within the nucleus. As dense chromatin regions decompact, they expand into chromatin-sparse areas and lead to an overall increase in nuclear volume. We attribute this increase in nuclear volume to chromatin decompaction. Similarly, changes in phase behavior arise from the homogenization caused by chromatin decompaction. Thus, we do not see a direct cause-and-effect relationship between nuclear volume and phase behavior or condensate dynamics; instead, both are consequences of chromatin decompaction and homogenization. Furthermore, we measure the real-time concentration of proteins using quantitative fluorescence microscopy, which remains unaffected by changes in nuclear volume, allowing us to accurately map the phase diagram.

There are also minor points that the authors might want to address:

Fig. 1g and Line 160: It would be helpful to have a line of slope 0.75 in Fig. 1g for reference.

We thank the reviewer for the good suggestion. We have added guideline of slope 0.75 in **Fig. 1(g)** and guidelines of slope 0.2 and slope 1 in **Fig. 1(f)**.

Fig. 1h: The binning, and particularly the representation of the histogram, is very problematic. First, the bins for large Δt overlap. For small Δt , there are huge gaps, which make interpreting this histogram difficult. Consequently, I am not convinced that there is a truly bimodal distribution. Generally, it might be better to use a cumulative distribution function, which does not require binning. There might also alternative representations of the data if the claim of bimodality is indeed true.

We agree that the presentation of the histogram is problematic, and appreciate this suggestion. We now replot the histogram and added the cumulative distribution function to better present the difference observed in control and TSA-treated cells. To demonstrate the bimodal distribution observed in control cells, we reduce the bin size to 0.001 to increase the visual resolution of the histogram, we also added an insert figure in **Fig. 1(h)** to show the zoom in view of the distribution in the bin range from 0 to 0.015.

Line 189: “[...] leading to an overall reduction in the mesh size of the chromatin network.” Do you have evidence that the mesh size changes? To me, this sounds as a hypothesis at this point and should probably be marked as such.

We agree that our description is not accurate, and we changed the sentence to reflect that it is an assumption. We do not directly measure the mesh size of the chromatin network, instead we indirectly infer the change in the mesh size by quantify the dynamics of GEM40 nanoparticles, whose MSD exhibits a downward shift with prolonged TSA treatment (24h and 36h) (**Supplementary Fig. S9(a)**), as well as decreased diffusion exponent and coefficient (**Supplementary Fig. S9(b) and (c)**).

Line 302: Why would the size “reach a plateau” when coarsening takes place?

We apologize for the use of the term "plateau," which may have been misleading. We have rephrased it to "a much slower growth stage" to avoid confusion. We agree that while the number of condensates reaches a plateau after the initial nucleation and growth stage, their size continues to increase over time, albeit at a much slower rate.

Line 302 and Fig. 3: It was claimed that the motion of the condensates are tracked for 1 minute, while in Fig. 3 the MSDs are only shown for 10 seconds. What is the reason for this discrepancy?

To address this concern, we reanalyzed the MSD for nearly two decades of lag time and replot **Fig. 3(b), (f) and (j)**.

Our initial intent was to use a smaller lag time to enhance the robustness of the measurement. As the time lag increases, fewer independent measurements are available to calculate the average displacement, leading to increased noise and uncertainty in the MSD values at longer times. It has been suggested the MSD is most robust when the maximum time lag is 20% of the total imaging time⁴³. Therefore, we initially chose 10s as the maximum lag time.

Line 325: Sentence “...we examined telomeres, which are protected ends of telomeres that assemble...” is confusing.

We have corrected the sentence. “To investigate if similar dynamics occur in an assembly attached to the chromatin network, we examined telomeres, which are protected ends of chromosomes that assemble into compact structures of a few hundred nanometers.”

Line 363: The label should be (k) and (l).

Thanks for pointing this out, we have corrected the figure legend.

Line 403 and Fig. 4: Why do you give the core-to-IDR ratios in powers of 2? Particularly in the figure, I find it difficult to get an intuitive grasp of numbers like 2^{-5} and 2^{-7} . I think powers of 10 are much more commonly used.

We appreciate this feedback. We agree that powers of 10 are more commonly used and generally easier to interpret. In our case, the core-to-IDR ratios range from 0.008 to 0.25. Using powers of 10 would only provide about 1.5 decades on the y-axis, which limits the resolution of our data representation. By using powers of 2, we can expand this range to 5 decades, allowing for a finer gradation and more detailed visualization of the data across the full range of ratios.

Fig. 4a: Is there a control study without TSA treatment showing that condensates persist? If not, one could claim that condensates disappear due to aging of the entire cell.

We thank the reviewer's suggestion and acknowledge that it's valuable to show examples of control cells to rule out other factors during the long-time cell culture. We performed a control experiment with no drug added and culture cell for 35h. We have observed no noticeable suppression of phase separation in cells, as shown in **Supplementary Fig. S14(a)**. We also tracked the same cell for continuous 35h, and observed no inhibition of phase separation, as shown in **Supplementary Fig. S14(b)**. In addition, we quantify the binodal phase boundary during the entire culture time, and observe no difference in the phase boundary, as shown in **Supplementary Fig. S15(a), (b) and (c)**. Therefore, we conclude the inhibition of phase separation is due to the TSA induced chromatin homogenization but less likely due to cell aging. We have added this point to the manuscript as well.

Fig. 4a,b: Is it possible to quantify the total amounts in the condensates and the dilute phase to check material conservation (e.g., using the data in panel b)? More generally, what is the role of production and degradation in these experiments?

Thank you for the question. Generally, the total material in a cell's nucleus is not conserved throughout its "lifespan" due to ongoing protein production and degradation. However, in our experiments, we minimize the impact of protein concentration fluctuations by measuring the real-time protein concentration in individual cells immediately before condensate formation through light activation. This approach ensures that we capture the true concentration of condensate proteins, accounting for any production and degradation processes. Importantly, the phase diagram representing the cell population is independent of the specific protein concentration in each individual cell during the time course of cell culture, as evidenced by the robust unchanged phase boundary observed during 32 hours of cell culture in control samples with no drug added (**Supplementary Fig. S15(a), (b) and (c)**). This indicates that our findings are robust and less likely to be affected by variations in protein concentration across different cells at different time points.

Line 432: What does "both before and after blue light activation" mean? Does it mean that some of the images in panel (a) are before blue light treatment? This can be clarified.

Thank you for catching this typo. The images show condensate formation after blue light activation, and the figure description has been corrected.

Line 557: A particularly interesting soft matter system is the follow-up study of ref.

42: <https://doi.org/10.1038/s41563-023-01703-0>. This study shows that elastic matrices can suppress coarsening (which might explain the slower coarsening rates observed in the TSA experiments). The accompanying theory paper <https://doi.org/10.1103/PhysRevX.14.021009> suggests that heterogeneity in elastic material properties could lead to such arrest, which sounds particularly relevant in the case of chromatin.

This is exciting, and we appreciate the suggested references. We agree that both references are highly relevant to our study, particularly the theoretical paper, which demonstrates that the heterogeneity of the mesh network can affect phase equilibrium. We have now included these references in our paper.

Fig. S2(b): It seems the size correlation coefficient divides into two groups for the control. Is there a reason?

Thanks for the observation. It is not entirely clear to us whether the correlation coefficient in the control group has split into two distinct groups. However, we did observe a broader distribution of correlation coefficients in the control group, likely due to the high degree of heterogeneity in chromatin density. This heterogeneity can introduce more noise and reduce the accuracy of fluorescence intensity measurements at the sites of individual condensates. In contrast, in the TSA-treated case, the chromatin network is spatially more homogeneous, allowing for more accurate measurement of chromatin intensity at the sites of individual condensates.

Reviewer #4 (Remarks to the Author):

1. Irianto, J. *et al.* Osmotic challenge drives rapid and reversible chromatin condensation in chondrocytes. *Biophys. J.* **104**, 759–769 (2013).
2. Martin, L. *et al.* A protocol to quantify chromatin compaction with confocal and super-resolution microscopy in cultured cells. *STAR Protoc.* **2**, 100865 (2021).
3. May, D. *et al.* Live imaging reveals chromatin compaction transitions and dynamic transcriptional bursting during stem cell differentiation in vivo. *Elife* **12**, (2023).
4. Ghosh, S. *et al.* Deformation microscopy for dynamic intracellular and intranuclear mapping of mechanics with high spatiotemporal resolution. *Cell Rep.* **27**, 1607-1620.e4 (2019).
5. Shin, Y. *et al.* Liquid Nuclear Condensates Mechanically Sense and Restructure the Genome. *Cell* **176**, 1518 (2019).

6. Tatton, N. A. & Rideout, H. J. Confocal microscopy as a tool to examine DNA fragmentation, chromatin condensation and other apoptotic changes in Parkinson's disease. *Parkinsonism Relat. Disord.* **5**, 179–186 (1999).
7. Hertzog, M. & Erdel, F. The Material Properties of the Cell Nucleus: A Matter of Scale. *Cells* **12**, (2023).
8. Noy, A. & Golestanian, R. Length scale dependence of DNA mechanical properties. *Phys. Rev. Lett.* **109**, 228101 (2012).
9. Wintner, O. *et al.* A Unified Linear Viscoelastic Model of the Cell Nucleus Defines the Mechanical Contributions of Lamins and Chromatin. *Adv. Sci.* **7**, 1901222 (2020).
10. Lammerding, J. Mechanics of the nucleus. *Compr. Physiol.* **1**, 783–807 (2011).
11. Cusack, M. *et al.* Distinct contributions of DNA methylation and histone acetylation to the genomic occupancy of transcription factors. *Genome Res.* **30**, 1393–1406 (2020).
12. Miranda, T. B. *et al.* DZNep is a global histone methylation inhibitor that reactivates developmental genes not silenced by DNA methylation. *Mol. Cancer Ther.* **8**, 1579–1588 (2009).
13. Xie, L. *et al.* 3D ATAC-PALM: super-resolution imaging of the accessible genome. *Nat. Methods* **17**, 430–436 (2020).
14. Arbach, H. E. *et al.* Chromatin accessibility analysis reveals distinct functions for HDAC and EZH2 activities in early appendage regeneration. *Wound Repair Regen.* **30**, 707–725 (2022).
15. Tóth, K. F. *et al.* Trichostatin A-induced histone acetylation causes decondensation of interphase chromatin. *J. Cell Sci.* **117**, 4277–4287 (2004).
16. Strickfaden, H. *et al.* Condensed chromatin behaves like a solid on the mesoscale in vitro and in living cells. *Cell* **183**, 1772-1784.e13 (2020).
17. Szórádi, T. *et al.* nucGEMs probe the biophysical properties of the nucleoplasm. *bioRxiv* 2021.11.18.469159 (2021) doi:10.1101/2021.11.18.469159.
18. Delarue, M. *et al.* mTORC1 Controls Phase Separation and the Biophysical Properties of the Cytoplasm by Tuning Crowding. *Cell* **174**, 338-349 e20 (2018).
19. Wirtz, D. Particle-tracking microrheology of living cells: principles and applications. *Annu. Rev. Biophys.* **38**, 301–326 (2009).

20. Van Hove, A. H., Wilson, B. D. & Benoit, D. S. W. Microwave-assisted functionalization of poly(ethylene glycol) and on-resin peptides for use in chain polymerizations and hydrogel formation. *J. Vis. Exp.* **80**, e50890 (2013).
21. Flory, P. J. & Rehner, J., Jr. Statistical mechanics of cross-linked polymer networks II. Swelling. *J. Chem. Phys.* **11**, 521–526 (1943).
22. Stephens, A. D., Banigan, E. J., Adam, S. A., Goldman, R. D. & Marko, J. F. Chromatin and lamin A determine two different mechanical response regimes of the cell nucleus. *Mol. Biol. Cell* **28**, 1984–1996 (2017).
23. Stephens, A. D., Banigan, E. J. & Marko, J. F. Chromatin’s physical properties shape the nucleus and its functions. *Curr. Opin. Cell Biol.* **58**, 76–84 (2019).
24. Wright, S. J. & Wright, D. J. Introduction to confocal microscopy. *Methods Cell Biol.* **70**, 1–85 (2002).
25. *Confocal Microscopy: Methods and Protocols.* (Springer, New York, NY, 2021).
26. Finn, E. H., Pegoraro, G., Shachar, S. & Misteli, T. Comparative analysis of 2D and 3D distance measurements to study spatial genome organization. *Methods* **123**, 47–55 (2017).
27. Waters, J. & Wittmann, T. *Quantitative Imaging in Cell Biology.* (Not Avail, 2014).
28. Identifying phase-separating biomolecular condensates in cells. *Nat. Chem.* **16**, 1050–1051 (2024).
29. Kapur, I., Boulier, E. L. & Francis, N. J. Regulation of Polyhomeotic condensates by intrinsically disordered sequences that affect chromatin binding. *Epigenomes* **6**, 40 (2022).
30. Keating, S. S., Bademosi, A. T., San Gil, R. & Walker, A. K. Aggregation-prone TDP-43 sequesters and drives pathological transitions of free nuclear TDP-43. *Cell. Mol. Life Sci.* **80**, 95 (2023).
31. Lee, D. S. W. *et al.* Size distributions of intracellular condensates reflect competition between coalescence and nucleation. *Nat. Phys.* **19**, 586–596 (2023).
32. Lee, D. S. W., Wingreen, N. S. & Brangwynne, C. P. Chromatin mechanics dictates subdiffusion and coarsening dynamics of embedded condensates. *Biophys. J.* **120**, 318a (2021).
33. Bracha, D. *et al.* Mapping Local and Global Liquid Phase Behavior in Living Cells Using Photo-Oligomerizable Seeds. *Cell* **175**, 1467–1480 e13 (2018).

34. Levensgood, J. D. & Tolbert, B. S. Idiosyncrasies of hnRNP A1-RNA recognition: Can binding mode influence function. *Semin. Cell Dev. Biol.* **86**, 150–161 (2019).
35. Loughlin, F. E. *et al.* The solution structure of FUS bound to RNA reveals a bipartite mode of RNA recognition with both sequence and shape specificity. *Mol. Cell* **73**, 490–504.e6 (2019).
36. McCreery, K. P. *et al.* Mechano-osmotic signals control chromatin state and fate transitions in pluripotent stem cells. *bioRxiv* 2024.09.07.611779 (2024) doi:10.1101/2024.09.07.611779.
37. Golkaram, M., Jang, J., Hellander, S., Kosik, K. S. & Petzold, L. R. The role of chromatin density in cell population heterogeneity during stem cell differentiation. *Sci. Rep.* **7**, 13307 (2017).
38. Heo, S.-J. *et al.* Aberrant chromatin reorganization in cells from diseased fibrous connective tissue in response to altered chemomechanical cues. *Nat Biomed Eng* **7**, 177–191 (2023).
39. Xu, X. *et al.* Chromatin remodeling and nucleoskeleton synergistically control osteogenic differentiation in different matrix stiffnesses. *Mater. Today Bio* **20**, 100661 (2023).
40. Hübner, B. *et al.* Ultrastructure and nuclear architecture of telomeric chromatin revealed by correlative light and electron microscopy. *Nucleic Acids Res.* **50**, 5047–5063 (2022).
41. Saxton, M. J. Diffusion of DNA-binding species in the nucleus: A transient anomalous subdiffusion model. *Biophys. J.* **118**, 2151–2167 (2020).
42. Amitai, A. & Holcman, D. Polymer physics of nuclear organization and function. *Phys. Rep.* **678**, 1–83 (2017).
43. Kepten, E., Weron, A., Sikora, G., Burnecki, K. & Garini, Y. Guidelines for the fitting of anomalous diffusion mean square displacement graphs from single particle tracking experiments. *PLoS One* **10**, e0117722 (2015).
44. Stephens, A. D. *et al.* Chromatin histone modifications and rigidity affect nuclear morphology independent of lamins. *Mol. Biol. Cell* **29**, 220–233 (2018).
45. Buchwalter, A., Kaneshiro, J. M. & Hetzer, M. W. Coaching from the sidelines: the nuclear periphery in genome regulation. *Nat. Rev. Genet.* **20**, 39–50 (2019).

We thank the reviewers for their thoughtful comments and critiques, which have significantly strengthened the manuscript. Below, we provide a point-by-point response.

Reviewers' comments:

Reviewer #1 (Remarks to the Author):

I acknowledge that the authors performed numerous additive experiments and analyses in an attempt to address the criticisms raised.

However, there are still relevant critical points that strongly limit the biological significance, novelty, and relevance of this manuscript.

Below are some critical points:

1. The simplistic, superficial, and biologically incorrect description of chromatin as a unified polymer that transits from an undefined heterogeneous to a homogeneous state represents the rationale of this work. Indeed, the authors state from the title that "Chromatin heterogeneity controls nuclear condensates dynamics and phase behavior." Chromatin is heterogeneous as its biophysical and biochemical state is highly and dynamically regulated, both at the nano and meso-scales. In addition, the 3D genome topology demarks specific spatial 3D chromatin compartments that respond to stimuli differently: some regions can sustain a large spectrum of perturbations, while others are more sensitive and responsive. In light of this, considering that chromatin will behave homogeneously in response to a single perturbation (TSA) is incorrect, and any conclusion based on this assumption is not sustainable. In addition, the proposed methodology used to support this statement (coefficient of variation of H2A signal) is arbitrary and not supported by any biological data.

We appreciate these comments. We believe there may be a misunderstanding regarding our claims. We did not make any claim regarding the dynamic response of chromatin to TSA. Instead, our study focuses on the outcome of TSA treatment, specifically that the chromatin network becomes spatially more homogeneous. This observation is consistent with findings reported in many prior studies, which have consistently demonstrated that TSA-induced acetylation leads to chromatin decompaction and redistribution, resulting in increased spatial homogeneity of chromatin organization.

Indeed, there are at least 8 independent studies showing that TSA treatment decreases the heterogeneity of the chromatin network. These studies have used a combination of genomics, imaging, and biophysical approaches, including ATAC-PALM, Image Correlation Spectroscopy (ICS), Spatially Resolved Scaling Analysis (SRSA), Fluorescence Correlation Spectroscopy (FCS), and Fluorescence Lifetime Imaging Microscopy (FLIM).

We have provided the references below along with direct quotes from each paper:

1. In Fejes Toth et al., *JCS* (2004)¹, authors reported that “The globally increased histone acetylation caused a reversible decondensation of dense chromatin regions and **led to a more homogeneous distribution.**” The authors further report that “Upon **TSA treatment** these dense regions vanished and a **more homogeneous chromatin distribution** was observed (Fig. 4A, C).”
2. In Görisch, S.M. et al., *JCS* (2005)², the authors report that “In interphase cells, the **TSA-induced histone acetylation** led to an increase of the chromatin correlation length l_c from $1.3 \pm 0.1 \mu\text{m}$ to $1.8 \pm 0.1 \mu\text{m}$. The value of l_c was determined from the autocorrelation of the DAPI images $G_1(r)$ and describes the size of chromatin substructures with similar density (Table 2, Fig. 3A). **Its increase reflects the disintegration of dense chromatin regions and the transition to a more homogeneous chromatin distribution** as described previously”
3. In Falk, M. et al., *Biochimica et Biophysica Acta (BBA)-Molecular Cell Research*(2008)³, the author reported that “Our observation of a **more homogenous distribution of chromatin in nuclei exposed to TSA** (Fig. 8C) and a slightly higher induction of DSBs in heterochromatic chromosomes (Fig. 6), is consistent with the observed TSA-induced relaxation of condensed chromatin domains containing several megabase pairs of DNA [28].”
4. In Casas-Delucchi et al., *NAR* (2012)⁴, the authors used a standard deviation of DAPI signal as a metric for DNA condensation: “mid-confocal sections of cells stained with DAPI were used to quantify the standard deviation of DAPI histograms, as a measure for the homogeneity of DNA compaction over the nucleus” and reported that “High resolution 3D-SIM images are presented to illustrate how the disruption of all three factors, histone hypoacetylation, H3K9m3 and DNA methylation, resulted in changes in the structural conformation of chromocenters, with **TSA having the most prominent effect, as seen by a more homogeneous DAPI staining**”.
5. In Wachsmuth, M. et al., *Epigenetics & Chromatin* (2016)⁵, the authors reported that “**After treatment of the cells with TSA**, chromatin became hyperacetylated and adopted a decondensed state of the chromatin fiber [70, 71]. This process resulted in a **homogeneous nuclear morphology and chromatin density distribution** (Fig. 4c).”
6. In Spagnol, S.T. et al., *PLoS One* (2016)⁶, the authors state that “**TSA treatment** resulted in a dramatic increase in the mean fluorescence lifetime relative to untreated controls which **indicated an increase in chromatin condensation state homogeneity** throughout the cell nucleus. ”
7. In Nozaki et al., *Mol Cell* (2017)⁷, the authors describe “Consistent with this notion, TSA-treated cells exhibited **more distributed nucleosome signals throughout** the nucleus than control cells.”

8. In Xie et al., Nat Methods (2020)⁸, the authors developed an imaging method adapted from ATAC-seq, and showed that “TSA treatment markedly **reduced $g(r)$** (Fig. 1d), suggesting **less clustering and a more homogenous distribution** of accessible chromatin in the nucleus.”

In conclusion, we believe these widely cited references collectively validate our experimental system by demonstrating that TSA treatment reduces the heterogeneity of the chromatin network, leading to increased spatial homogeneity. We have now included these citations in the manuscript to provide additional context and support for our findings.

2. The strong limitation of the described approach is also highlighted by the observation by which the motion of the condensates (both the synthetic and the endogenous one) resulted in being reduced in a condition of a “homogeneous” chromatin state. These results are interpreted as the resulting spatial distribution of chromatin reduces the available space for condensation motion. However, this is mere speculation: the obtained measurements are recoding the different motions of condensates upon long-term TSA treatment, which causes many indirect effects that are not controlled nor properly considered in the interpretation of the results. Another critical point: Are these differences in mobility biologically relevant?

In the manuscript, we qualitatively compare changes in the apparent nuclear mechanics before and after TSA treatment by analyzing the dynamics of nanoparticles, which is a widely used approach to characterize the mechanical properties of the intracellular environment^{9–11}. Specifically, we used the experimental method developed by the lab of Liam Holt¹⁰. This analysis relies on quantifying the dynamics of nanosized GEM40 nanoparticles, where reduced motion indicates increased stiffness in the nuclear interior (**Fig. 1(i)-(l)**). GEM40 nanoparticles are externally engineered and introduced into cells. As TSA is a histone deacetylase inhibitor (HDACi) that acts on histones, it is unlikely to directly affect GEM40 particles. Therefore, the GEM40 measurements reflect nuclear mechanics independent of TSA treatment.

We believe that the differences in mobility are biologically relevant; however, we underscore that assessing particular biological/functional relevance is not the focus of the present manuscript. Indeed, biomolecular condensates play critical roles in organizing nuclear processes, such as transcription and RNA processing, by compartmentalizing and concentrating specific biomolecules to enhance their interactions and efficiency¹². Their mobility can potentially influence their ability to interact with specific genomic regions or other nuclear components, which is essential for regulating gene expression and maintaining nuclear organization^{13–15}. We acknowledge that further studies are needed to directly link these mobility changes to specific biological outcomes, such as transcriptional regulation or stress response, as condensates also play key roles in cellular adaptation to environmental changes.

3. Some incongruency in the obtained results strongly limits the significance of the robustness of the performed analyses. For example, how is it possible that the Cajal bodies (whose mobility has been determined by Coilin labeling) would show a “Brownian motion” with a diffusion

exponent alpha of 1? This measurement is inconsistent with many other MSD data referring to chromatin factors, TFs, and condensates (including the data presented in this work), considering that chromatin would constrain the motion of macromolecules within the nuclear space.

We thank the reviewer for pointing out that chromatin dynamics within the nuclear space can impose constraints on the motion of nuclear bodies and macromolecules, leading to a diffusion exponent alpha between 0 and 1. However, our result is consistent with the evidence in the literature supporting the possibility of Cajal bodies displaying diffusive exponents (α) around 1 (or even great than), which is indicative of motion resembling free diffusion (or driven motion, in the case of $\alpha > 1$). For instance, Görisch et al, PNAS (2004) observe $\alpha \geq 1$ in their studies¹⁶. Similarly, Platani et al, Nature Cell Biol (2002) report a diffusive exponent $\alpha > 1$ for Cajal bodies in living cells (Figure 5, blue line)¹⁷. These findings corroborate that such motion is physiologically plausible and consistent with our results.

4. These aspects are further underlined by the poor statistics applied to determine the correctness of the proposed null hypothesis in each experiment. Indeed, many experiments were performed on two biological replicates. In addition, in all the experiments, the applied statistical tests consider “n” the number of objects measured/condition and not the number of biological replicates from which the values have been retrieved. By doing so, the statistical tests do not address the null hypothesis by which the measured averaged difference among the tested conditions depends on the treatment (TSA). However, the authors drew their conclusions based on the mentioned null hypothesis, which has not been tested statistically. The reported issues are well exemplified by the statistical test applied in Fig. 2e and 2f, in which the very marginal differences in the artificial biomolecular condensates diameters (Corelet system) results being statistically significant only because the authors used as “n” the number of condensates (>500) retrieved from only two biological replicates.

We appreciate the thoughtful critique of our statistical analyses. In response, we have now included at least three biological replicates for all major results in the manuscript to ensure statistical rigor. Specifically, we re-calculated statistical significance such as in Fig. 2e and 2f, as well as SI Fig. S10 and Fig. S11 by calculating cell-averaged condensate size, and considering n as the number of cells (instead of the number of condensates before). We still observe a strong statistical significance with the much-lowered sample size of cell-average measurements, further proving that the change associated with TSA perturbation is robust and statistically significant. Additionally, the robustness of our findings is further supported by the high reproducibility of the results across multiple cell lines, a range of chemical perturbations, and condensates with various intrinsically disordered regions (IDRs).

5. If someone would consider the correlation between transition in chromatin state (from heterogeneous to homogeneous) and condensates formation still biologically relevant (regardless of the biological inconsistencies described in point 1), then this topic will result in

poor novelty, as the same group published different papers covering the same topic, including the very recent publication entitled “Chromatin compaction during confined cell migration induces and reshapes nuclear condensates”.

We would like to clarify that both the topic and findings of Zhao et al., Nat Comms 2024 are very distinct from this paper, Zhao et al. examine the fusion, fission, and formation of condensates during confined cancer cell migration. In contrast, our study provides a general framework for understanding how chromatin's physical properties regulate nuclear condensate behavior (formation, mobility, size and coarsening dynamics etc.). Moreover, the robustness of our findings is demonstrated through reproducibility across multiple cell lines, diverse chemical perturbations, and condensates with various intrinsically disordered regions (IDRs).

Given the major biological issues, the inconsistency of the data, and the inappropriate statistical approaches, I do not support the publication of this manuscript in Nature Communications.

Reviewer #2 (Remarks to the Author):

The authors have addressed most of my comments except the following:

1) For the endogenous condensates: it is understandable that not all biophysical quantifications can be carried out with endogenous condensates. I suggest the authors be more specific in drawing conclusions and avoid extrapolating observations made with engineered condensates to endogenous condensates. For example, the authors state “We demonstrate that decreasing chromatin heterogeneity with epigenetic modifying drugs impairs condensate growth and mobility by shifting the binodal phase boundary.” The shifting in phase boundary was not demonstrated with endogenous condensates. Also, for the nuclear speckle sizes added in Figure 2jk, unlike the engineered system, nuclear speckle formation is known to be linked to transcription. Therefore, an alternative interpretation of the size changes after TSA is because of its effect on transcription. Another way to change chromatin heterogeneity, such as that shown in Figure R2, would be needed to confirm the interpretation.

Thank you for the comment. We agree that the statement needed refinement. We have now revised the statement in the abstract to be: ‘We demonstrate that decreasing chromatin heterogeneity with epigenetic modifying drugs decreases the mobility of both endogenous and engineered condensates, impairs condensate growth by shifting the binodal phase boundary of engineered condensate.’

We also appreciate the reviewer’s insightful suggestion to explore additional methods for altering chromatin heterogeneity, such as those shown in previous Figure R2. In response, we have performed the suggested experiment and included the results in the revised manuscript as Figure S18. We believe this addition strengthens our findings and provides further evidence supporting our conclusions.

2) For the cell cycle and nuclear volume after TSA: the effect of the cell cycle on chromatin heterogeneity was addressed with new data, but the dependence of phase boundary, condensate sizes, and mobility on the cell cycle was not measured. For the volume effect, this would not be a concern for the engineered condensates where phase diagrams are mapped, but for the endogenous condensates such as newly added Figure 2jk, lowering of protein concentration due to increased nuclear volume would play a role, and this needs to be considered. Measuring eYFP-SRRM2 intensity or protein level with Western may not be sufficient as nuclear speckles, like many endogenous condensates and unlike the engineered condensates, are formed by multiple components. Again, I think perturbing the chromatin heterogeneity using a different method, such as that shown in Figure R2, would be needed to confirm that the results can be attributed to chromatin heterogeneity.

Thank you for this valuable suggestion. We agree that perturbing chromatin heterogeneity using an alternative method would provide additional evidence. As recommended by the reviewer, we performed the same experiment as shown in the previous Figure R2. We have now included the results in Figure S18. We believe this addition enhances our findings and provides further evidence to support our conclusions.

3) A drug to increase chromatin heterogeneity: the data in Figure R2 is very convincing and would address all concerns regarding the side effects of TSA outlined in points 1 and 2 above. Interestingly, the authors decide not to include the data or the method in this paper. Without such a confirmation, the side effects of TSA remain a concern for this reviewer.

Thank you for your insightful comments and for suggesting adding the data presented in Figure R2. We agree that this data addresses concerns regarding the potential side effects of TSA and provides further evidence supporting the role of chromatin heterogeneity in condensate behavior.

We didn't include this data initially in the manuscript because our primary focus was on using TSA or DZNep to reduce chromatin heterogeneity. However, we now recognize that incorporating this additional data strengthens our conclusions and directly addresses concerns about TSA's side effects. In response, we have conducted new experiments as outlined in Figure R2 and incorporated the results into the manuscript as Figure S18. Additionally, we have included a detailed description of the methodology in the Materials and Methods section.

Reviewer #3 (Remarks to the Author):

The revised manuscript is much improved and the authors addressed almost all comments satisfactorily. There are only a few remaining points:

1) The scalings in Fig 3 are much improved, but the lines in panels b,f, j are still not straight enough to have conclusive scaling laws in my opinion. I accept that they are sufficient for the current conclusion of the manuscript, but it might be advisable to tone down the claims about scaling laws.

Thank you for the comments. We acknowledge that, strictly speaking, the MSD does not exhibit a perfectly power law across the entire time scale. The curve was fitted to a power-law model solely to enable a preliminary comparison of Corelet dynamics before and after TSA treatment. In accordance with the reviewers' suggestions, we have toned down the claims.

2) Regarding tracking the condensates, the authors explain that there are technical difficulties for direct observation of coarsening events. However, two different mechanisms should be already distinguishable from the moving paths. Since the MSDs are calculated, the moving paths of condensates must be already obtained by tracking the positions of the condensates (e.g. in Fig. 3 a,e,j). It should be possible to check whether the condensates coarsen through Ostwald ripening, where the condensates's positions stay almost unchanged, or through Brownian motion, where the the paths of two condensates may meet and merge into one.

We thank the reviewer for their insightful suggestion. The MSD analysis is indeed based on tracking the positions of condensates, and these trajectories have been obtained. Previous studies have demonstrated that the dominant growth mechanism for nuclear condensates occurs through coalescence events. While coalescence events can be readily identified by observing the merging of condensates, as shown in the included supplementary Movie S1, directly observing Ostwald ripening remains challenging. This is because Ostwald ripening involves growth or shrinkage without significant movement of condensates, which is almost impossible in living cells. Additionally, the surface tension of intracellular condensates is low, on the order of 10^{-6} N/m (Caragine et al, (PRL, 2018))¹⁸, which makes Ostwald ripening less likely to be dominant. We agree that trajectory analysis could provide additional insights; however, a rigorous distinction between these mechanisms would require higher temporal and spatial resolution, as well as tracking individual condensates in 3D, to enable a more detailed characterization of their trajectories and sizes.

3) The authors might want to check the Journal information of Ref. 42, which seems to be incorrect.

Thank you for the comments. We have updated and corrected the citation details for Reference 42.

4) The response to the concern of using powers of 2 in the labeling of ticks in Fig. 4 is confusing. The authors claim that "By using powers of 2, [they] can expand this range to 5

decades”, which is clearly wrong, since there are still only 1.5 decades independent of the base used for the labeling. It is likely much more intuitive to label powers of 10 (even when using $10^{1.5}$ or something like that). Even better might be simple numbers or a linear axis, which should not change the presentation much, given that the logarithmic axis covers less than two decades.

Thank you for the suggestion. We have updated the y-axis to a logarithmic scale with base 10 for all figures in the manuscript and supplementary information.

Reviewer #4 (Remarks to the Author):

1. Tóth, K. F. *et al.* Trichostatin A-induced histone acetylation causes decondensation of interphase chromatin. *J. Cell Sci.* **117**, 4277–4287 (2004).
2. Görisch, S. M., Wachsmuth, M., Tóth, K. F., Lichter, P. & Rippe, K. Histone acetylation increases chromatin accessibility. *J. Cell Sci.* **118**, 5825–5834 (2005).
3. Falk, M., Lukášová, E. & Kozubek, S. Chromatin structure influences the sensitivity of DNA to gamma-radiation. *Biochim. Biophys. Acta* **1783**, 2398–2414 (2008).
4. Casas-Delucchi, C. S. *et al.* Histone hypoacetylation is required to maintain late replication timing of constitutive heterochromatin. *Nucleic Acids Res.* **40**, 159–169 (2012).
5. Wachsmuth, M., Knoch, T. A. & Rippe, K. Dynamic properties of independent chromatin domains measured by correlation spectroscopy in living cells. *Epigenetics Chromatin* **9**, 57 (2016).
6. Spagnol, S. T. & Dahl, K. N. Spatially resolved quantification of chromatin condensation through differential local rheology in cell nuclei fluorescence lifetime imaging. *PLoS One* **11**, e0146244 (2016).
7. Nozaki, T. *et al.* Dynamic organization of chromatin domains revealed by super-resolution live-cell imaging. *Mol. Cell* **67**, 282-293.e7 (2017).
8. Xie, L. *et al.* 3D ATAC-PALM: super-resolution imaging of the accessible genome. *Nat. Methods* **17**, 430–436 (2020).
9. Lee, J. S. H. *et al.* Nuclear lamin A/C deficiency induces defects in cell mechanics, polarization, and migration. *Biophys. J.* **93**, 2542–2552 (2007).

10. Szórádi, T. *et al.* nucGEMs probe the biophysical properties of the nucleoplasm. *bioRxiv* 2021.11.18.469159 (2021) doi:10.1101/2021.11.18.469159.
11. McLaughlin, G. A. *et al.* Spatial heterogeneity of the cytosol revealed by machine learning-based 3D particle tracking. *Mol. Biol. Cell* **31**, 1498–1511 (2020).
12. Niu, X. *et al.* Biomolecular condensates: Formation mechanisms, biological functions, and therapeutic targets. *MedComm* **4**, e223 (2023).
13. Li, W. & Jiang, H. Nuclear protein condensates and their properties in regulation of gene expression. *J. Mol. Biol.* **434**, 167151 (2022).
14. Schede, H. H., Natarajan, P., Chakraborty, A. K. & Shrinivas, K. Organization and regulation of nuclear condensates by gene activity. *bioRxiv* 2022.09.19.508534 (2022) doi:10.1101/2022.09.19.508534.
15. Negri, M. L., D’Annunzio, S., Vitali, G. & Zippo, A. May the force be with you: Nuclear condensates function beyond transcription control: Potential nongenetic functions of nuclear condensates in physiological and pathological conditions. *Bioessays* **45**, e2300075 (2023).
16. Görisch, S. M. *et al.* Nuclear body movement is determined by chromatin accessibility and dynamics. *Proc. Natl. Acad. Sci. U. S. A.* **101**, 13221–13226 (2004).
17. Platani, M., Goldberg, I., Lamond, A. I. & Swedlow, J. R. Cajal body dynamics and association with chromatin are ATP-dependent. *Nat. Cell Biol.* **4**, 502–508 (2002).
18. Caragine, C. M., Haley, S. C. & Zidovska, A. Nucleolar dynamics and interactions with nucleoplasm in living cells. *Elife* **8**, (2019).

Response to the reviewers' comments

We would like to sincerely thank the reviewers and the editor for their careful evaluation of our manuscript and for the constructive feedback provided. We appreciate the thoughtful comments and suggestions, which have helped us to clarify and strengthen our work. To address this last round of comments, we have added new experiments and data, and have revised the manuscript accordingly. Below, we provide a detailed, point-by-point response to each reviewer's comment, indicating the changes made in the revised manuscript.

Reviewer #2 (Remarks to the Author)

The added sorbitol data have already been published in their previous paper. Interestingly, the authors repeated the experiment here instead of referring to the published paper. Since the point was made in the previous paper, this paper has little significance in that regard. The effect on endogenous condensates is new, but the sorbitol experiment was not done on Cajal bodies or nuclear speckles.

We thank Reviewer 2 for their careful reading of this and our prior papers. While the previous paper on the role of constrained migration in the intracellular phase behavior demonstrated the effect of sorbitol on the phase diagram using MDA-MB-231 cells, it was unclear whether these findings would extend to U2OS cells, which is the cell line used in the current study; indeed, the phase boundary and potential sorbitol dependence mapped by the Corelets system could vary across different cell lines. To address this, we repeated the sorbitol experiments in U2OS cells and found that increased chromatin heterogeneity induced by sorbitol promotes phase separation in this context as well. We trust that the reviewer agrees that confirming this effect in U2OS cells is an important control experiment, and adds value to the present manuscript.

We agree with the reviewer that assessing the effect of sorbitol on endogenous condensates would further strengthen our findings. In response, we performed additional experiments using U2OS cells with fluorescently tagged Cajal bodies. Our results show that sorbitol-induced chromatin heterogeneity leads to an increase in Cajal body size, as shown in **Supplementary Fig. S13(a), (b) and (c)**.

Reviewer #3 (Remarks to the Author)

The authors addressed my specific comments satisfactorily. Concerning the comments of the other referees, I agree with them and think that the authors might over-interpret their results. As far as I can see, the authors essentially showed a correlation between chromatin heterogeneity and the behavior of condensates, but the title and abstract imply causation. It is unclear from the provided data whether heterogeneity or any other aspect that might be changed by the treatment (e.g., overall stiffness or chemical properties) is responsible for the observed behavior. However, the observed correlation is definitely interesting and points toward a cross-talk between condensates and their environment. I thus believe this work should be published if causation is no longer implied or supporting evidence for it is added.

We thank Reviewer 3 for their constructive feedback. We agree that our data establish a correlation between chromatin heterogeneity and condensate behavior, and while the causal relationship between the two is likely, and indeed an explanation that makes good physical sense, the direct causal relationship has not been formally shown (although we note that formally demonstrating causality of a single variable is always nearly impossible). Accordingly, we have revised relevant sections throughout the manuscript, including qualifiers to clarify this. We appreciate the reviewer's suggestion and believe these changes improve the clarity and rigor of our manuscript.